# Trade-offs shaping transmission of sylvatic dengue and Zika viruses in monkey hosts

Kathryn A. Hanley ®[1,8] ✉, Hélène Cecilia ®[1,8], Sasha R. Azar ®[2,3], Brett A. Moehn ®[1], Jordan T. Gass ®[1], Natalia I. Oliveira da Silva ®[2], Wanqin Yu ®[1], Ruimei Yun ®[2], Benjamin M. Althouse[1,4], Nikos Vasilakis ®[2,5,6] & Shannan L. Rossi ®[2,5,6,7]

Mosquito-borne dengue (DENV) and Zika (ZIKV) viruses originated in Old World sylvatic (forest) cycles involving monkeys and canopy-living *Aedes* mosquitoes. Both viruses spilled over into human transmission and were translocated to the Americas, opening a path for spillback into Neotropical sylvatic cycles. Studies of the trade-offs that shape within-host dynamics and transmission of these viruses are lacking, hampering efforts to predict spillover and spillback. We infected a native, Asian host species (cynomolgus macaque) and a novel, American host species (squirrel monkey) with sylvatic strains of DENV-2 or ZIKV via mosquito bite. We then monitored aspects of viral replication (viremia), innate and adaptive immune response (natural killer (NK) cells and neutralizing antibodies, respectively), and transmission to mosquitoes. In both hosts, ZIKV reached high titers that translated into high transmission to mosquitoes; in contrast DENV-2 replicated to low levels and, unexpectedly, transmission occurred only when serum viremia was below or near the limit of detection. Our data reveal evidence of an immunologically-mediated trade-off between duration and magnitude of virus replication, as higher peak ZIKV titers are associated with shorter durations of viremia, and higher NK cell levels are associated with lower peak ZIKV titers and lower anti-DENV-2 antibody levels. Furthermore, patterns of transmission of each virus from a Neotropical monkey suggest that ZIKV has greater potential than DENV-2 to establish a sylvatic transmission cycle in the Americas.

Spillover of zoonotic, arthropod-borne viruses (arboviruses) into humans is accelerating, with outcomes ranging from dead-end infections to pandemics[1]. Because of the unparalleled mobility of humans, these viruses are being transported across the world, creating the potential for spillback into novel wildlife reservoirs[2]. Such

translocation is not new: yellow fever virus (YFV), which originated in a sylvatic cycle in the forests of Africa involving non-human primates (NHPs) and arboreal mosquitoes, was carried via sailing ships to the neotropics centuries ago[3,4]. Soon after, YFV established urban transmission in the Americas and spilled back into a sylvatic cycle

[1]Department of Biology, New Mexico State University, Las Cruces, NM 88003, USA. [2]Department of Pathology, University of Texas Medical Branch, Galveston, TX 77555, USA. [3]Center for Tissue Engineering, Department of Surgery, Houston Methodist Research Institute, Houston Methodist Hospital, Houston, TX 77030, USA. [4]Information School, University of Washington, Seattle, WA 98105, USA. [5]Center for Vector-Borne and Zoonotic Diseases, University of Texas Medical Branch, Galveston, TX 77555, USA. [6]Institute for Human Infection and Immunity, University of Texas Medical Branch, Galveston, TX 77555, USA. [7]Department of Microbiology and Immunology, University of Texas Medical Branch, Galveston, TX 77555, USA. [8]These authors contributed equally: Kathryn A. Hanley, Hélène Cecilia. ✉e-mail: khanley@nmsu.edu

maintained in neotropical primates and mosquitoes that plagues South America to present day[3,4]. The four serotypes of dengue virus (DENV-1-4) underwent a similar journey, albeit their ancestral sylvatic cycles occur in Asia, but compelling evidence of a sylvatic cycle of DENV in the Americas has never been detected despite being actively sought[5-7]. Thus, when ZIKV was detected in the Americas in 2015, there was great concern, but equally great uncertainty, about its potential to establish a neotropical sylvatic cycle[8-10]. While multiple instances of ZIKV infection in neotropical primates have now been documented[11,12], the potential for such spillback events to launch a sylvatic cycle remains unclear. Furthermore, the sylvatic cycles of DENV and ZIKV persist in Asia and Africa, where both continue to spill over into humans and are sometimes translocated to new continents, e.g.[13-19].

Mathematical models using within-host viral dynamics to predict between-host transmission are key for predicting arbovirus spillover, spread, and spillback[20]. These models assume within-host trade-offs that influence virus transmission[21], particularly the trade-off between instantaneous pathogen transmission and infection duration[22,23]. Host mortality is often invoked in theoretical studies as the mechanism that regulates infection duration, however most pathogens are cleared by the immune response. Indeed, for sylvatic DENV and ZIKV there is no evidence of disease in infected Old World monkeys[24-26], although sylvatic ZIKV was first isolated in Uganda from a febrile sentinel rhesus macaque, an Asian species imported to Africa for viral surveillance[16], and poses similar risks to the developing fetus in Old World monkeys as do human-endemic lineages of the virus[27-29]. Thus, transmission of these viruses likely depends on the trade-off between the magnitude of viral replication and immune clearance[30].

Studies of replication-clearance trade-offs in arboviruses are scarce. The majority of publications on the intra-host dynamics of arboviruses during experimental infection of vertebrate reservoir hosts found an inverse relationship between the magnitude (peak titer) of infection and duration of infection[31,32]. However, most of these studies delivered virus to the host via needle injection, which differs from natural transmission via mosquito bite[33-35]. We formulated a dynamical model comparing the effect of a "tortoise" strategy of low magnitude, long-duration viremia and a "hare" strategy of short-duration, high-magnitude viremia and showed that arboviruses adopting a tortoise strategy had higher rates of persistence in both host and vector populations[32]. Ben-Shachar and Koelle[36] investigated the replication-clearance trade-off in DENV using within-host simulation models based on data from a human cohort study in which mosquitoes fed on infected individuals, and found that a replication-clearance trade-off selects for intermediate DENV virulence. Importantly for the current study, they invoked natural killer (NK) cells to represent the innate immune response driving this effect.

In the current study, we investigated trade-offs between replication and clearance of sylvatic DENV and ZIKV, as well as specific immune responses associated with these trade-offs and potential differences in trade-offs between native and novel NHP hosts. This study tested three a priori hypotheses, of which the first is fundamental and the latter two consider mechanism and evolutionary context, respectively: Hypothesis 1 - sylvatic arboviruses experience a replication-clearance trade-off in both native and novel hosts, as evidenced by a positive relationship between the dose of virus delivered and peak virus titer, a positive relationship between virus titer and transmission to mosquitoes, and a negative relationship between peak virus titer and duration of viremia; Hypothesis 2 - NK cells can serve as a proxy for the innate immune responses that shape this trade-off, as evidenced by a negative relationship between NK cell mobilization immediately post-infection (pi) and peak virus titer as well as levels of neutralizing antibody, and Hypothesis 3 - sylvatic arboviruses have achieved an optimal replication-clearance trade-offs in native hosts but have not had evolutionary opportunity to reach this optimum in novel hosts, resulting in less transmission from novel hosts.

## Results

### Overview of experimental design

The putative native host in this study was the cynomolgus macaque, *Macaca fascicularis*. Rudnick and colleagues isolated sylvatic DENV from sentinel cynomolgus macaques held in the forest canopy in Malaysia and also showed that ≥ 50% of wild, forest-living cynomolgus macaques in Malaysia were seropositive for at least one serotype of DENV[37-39]. Other studies in Asia have also detected seropositive cynomolgus macaques (reviewed in ref. 40). The role of NHPs as hosts for ZIKV in Asia, and indeed the existence of an Asian sylvatic ZIKV cycle, is murkier because it is less studied. Wikan and Smith[41] have proposed that the sparsity of human ZIKV infections is Asia is consistent with spillover from a sylvatic cycle. Of 234 wild cynomolgus macaques tested in Malaysia[42], 1.3% were seropositive for ZIKV, and, in a singular case, an Australian traveler became infected with ZIKV after being bitten by a cynomolgus macaque in Indonesia[43]. Moreover, machine learning models have pinpointed cynomolgus macaques as having the highest risk of all primate species worldwide to serve as a host for ZIKV[44].

The novel host was the neotropical squirrel monkey, *Saimiri boliviensis boliviensis*, which machine learning models have identified as a high risk for spillback of ZIKV[44]. The two sylvatic viruses were DENV-2 strain P8-1407 and ZIKV strain DakAR 41525; the former was isolated from a sentinel cynomolgus macaque in the forest canopy in Malaysia[37-39] and the latter was isolated from an *Aedes africanus*, a primatophilic mosquito species that is generally confined to forest habitats[45], in Senegal. Virus was delivered by infected *Ae. albopictus* mosquitoes, which are native to Asia but distributed across much of the world, a likely bridge vector for spillover and spillback of both DENV and ZIKV[46], and a vector in outbreaks of sylvatic ZIKV in Gabon[47].

Figure 1 shows the experimental design for this study. To infect macaques, cartons of either 1 (low dose, $n = 5$ monkeys) or 10 (high dose, $n = 5$ monkeys) *Ae. albopictus* that had been intrathoracically (IT) inoculated with a sylvatic Malaysian strain of DENV-2, or 15 (high dose, $n = 3$ monkeys) *Ae. albopictus* IT inoculated with a sylvatic African strain of ZIKV, or 10 uninoculated, control *Ae. albopictus* ($n = 3$ monkeys) were placed upon a monkey's ear for approximately 10 minutes. Engorged mosquitoes were separated, incubated for 2 days, and force-salivated to estimate dose of virus delivered. Squirrel monkeys were infected with either a high dose (15 mosquitoes) of a sylvatic Malaysian strain of DENV-2 ($n = 10$ monkeys) or a sylvatic African ZIKV strain ($n = 10$ monkeys) or control mosquitoes ($n = 4$ monkeys). Because the small size of squirrel monkeys constrained blood draw frequency, monkeys were assigned to one of two cohorts and bled prior to infection and on alternating days in the first week pi. NK cells, neutralizing antibody titers via plaque reduction neutralization titers (PRNTs), weight and temperature were measured on designated days prior to and post-infection (dpi). On designated dpi, uninfected *Ae. albopictus* were allowed to feed on each animal to monitor transmission.

All raw data from macaques and squirrel monkeys infected with DENV-2 or ZIKV, save for temperature and PRNT values against non-infecting viruses, are provided in Supplementary Datasets 1, 2, 3, and 4, respectively. Following infection with either DENV-2 or ZIKV, body temperatures of macaques[48,49] and squirrel monkeys[50] stayed within the normal range of each species (Fig. S1) and there were no differences among treatment groups, nor were there any differences among treatment groups in changes in body weight change (Fig. S2).

### Sylvatic DENV-2 delivered by mosquito bite to both macaques and squirrel monkeys resulted in nearly 100% seroconversion with high neutralizing antibody titers

In IT-inoculated mosquitoes used to deliver DENV-2 to macaques, virus was not initially detectable via serial dilution and immunostaining of saliva samples in monolayers of mosquito cells. After saliva were subjected to one blind passage in mosquito cells, the virus was detected in a subset of salivations (Table 1). Thus, we used the number of mosquitoes

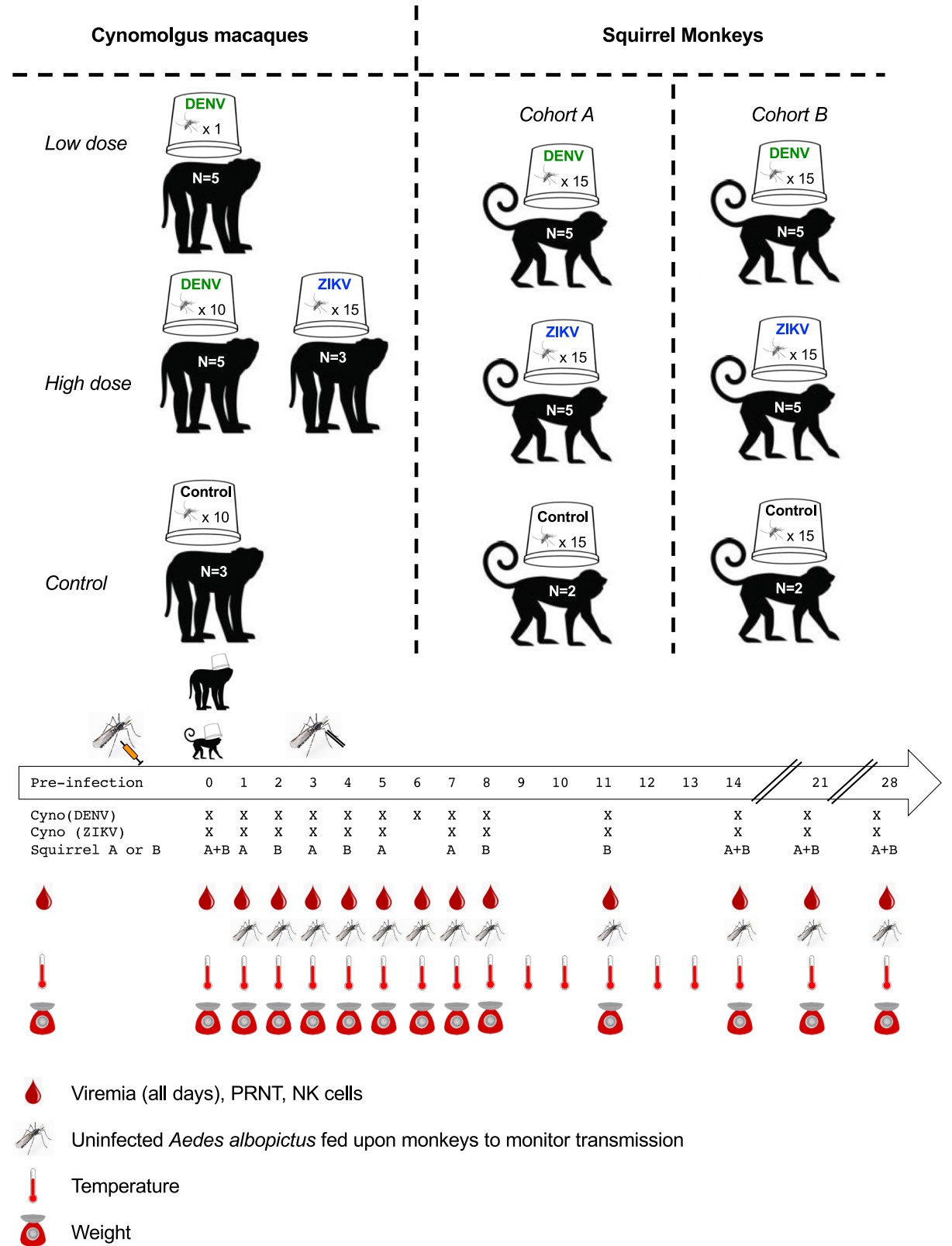

salivating detectable virus as a proxy for the total dose of DENV delivered to macaques. For macaque, UG253A, the single mosquito that fed produced no detectable virus in the saliva, and UG253A never produced detectable viremia or seroconverted (defined here as a PRNT50 titer, the most permissive cutoff, ≥ 20; Supplementary Dataset 5). Hence UG253A was reassigned, *post hoc*, to the control treatment. Macaque SB393 received a single bite from a mosquito that did not salivate

detectable virus, but this macaque seroconverted, demonstrating that it had become infected. Of the remaining eight macaques in the low and high-dose treatments, seven seroconverted, but one, BC407, did not, even though it received a bite from a mosquito with detectable virus in the saliva (Table 1). All statistical analyses of neutralizing antibody levels were conducted using PRNT80 values, a more stringent measure consistent with our previous study of sylvatic DENV-2 in African green

**Fig. 1 | Overview of experimental infections of cynomolgus macaques and squirrel monkeys with dengue virus serotype 2 (DENV-2, in green) and Zika virus (ZIKV, in blue) and subsequent sampling.** Blood samples were used to measure viremia, natural killer (NK) cells, and neutralizing antibody as plaque reduction neutralization titer (PRNT). Monkey body temperature was monitored continuously via implanted transponder and weight was monitored on designated days. Infection and monitoring were similar in macaques and squirrel monkeys, save that: (i) the sample size of infected and uninfected mosquitoes was increased from 10 in DENV-2 infected and control macaques to 15 in ZIKV-infected macaques and all squirrel monkeys, (ii) stool was collected daily from macaques to screen for occult blood; stool was not collected from squirrel monkeys due to pair housing of animals, (iii) mosquitoes that fed on infected squirrel monkeys on dpi 3 and 4 were force-salivated after a 14 day incubation, but this was not done in macaques, and (iv) squirrel monkeys were euthanized at the conclusion of the experiment and necropsies were conducted; euthanasia was not performed on macaques. In one instance, blood sampling (dpi 7) and mosquito feeding (dpi 8) occurred on different days for macaques. Images used under license from Shutterstock.com.

monkeys[24]. The harmonic mean PRNT80 value (values <limit of detection (LOD) set at 19) for macaques was 69.7 (Table 1, Fig. 2A) Of the mosquitoes used to deliver DENV-2 to squirrel monkeys, the virus was detectable in the saliva of 30/45 IT inoculated mosquitoes by titer of collected saliva or passaged saliva (Table 2). All squirrel monkeys seroconverted (PRNT50 > LOD), with a harmonic mean PRNT80 value of 91.4 (Table 2, Fig. 2B). There were no notable findings in necropsy reports for squirrel monkeys infected with DENV-2.

### Sylvatic DENV-2 replicated to low levels in macaques and squirrel monkeys and showed unexpected patterns of transmission

We initially predicted, based on hypothesis 1, that increasing dose of virus would lead to increasing peak viremia, which in turn would be associated with greater instantaneous transmission to mosquitoes but a shorter duration of viremia. It was not possible to test most of these predictions in macaques infected with DENV-2, as viremia was detected from serum via cell culture-based assay in only three macaques, one from the low dose and two from the high dose treatment (Table 1, Fig. 3A). Furthermore, only three macaques transmitted DENV-2 to mosquitoes, all from the high dose treatment (Table 1, Fig. 3A, Supplementary Dataset 6). Intriguingly, virus was not detected from raw or passaged serum in any macaque on the day of transmission to mosquitoes or, in the case of macaques FR840 and BC167, at any point during the study. It was possible to analyze the number of infectious bites received and the likelihood of becoming viremic, as evidenced by detectable virus in raw or passaged serum or mosquitoes ($n = 5$ viremic macaques total). Consistent with Hypothesis 1, detectably viremic animals received a significantly greater number of infectious bites than animals that were not detectably viremic (logistic regression, DF = 1, n = 9, $\chi^2 = 4.63$, P = 0.03).

Only three squirrel monkeys produced viremia that could be detected from serum (Table 2, Fig. 3B). Two monkeys that produced viremia detectable in serum transmitted DENV-2 to mosquitoes, but only when their viremia was at its lowest point. Another four monkeys that did not produce detectable viremia nonetheless transmitted DENV-2 to mosquitoes (Table 2, Fig. 3B). Notably, for mosquitoes fed on both macaques and squirrel monkeys, DENV-2 could be detected more often in legs than in bodies (Supplementary Dataset 6). Contra Hypothesis 1, dose of DENV-2 delivered had no significant impact on the likelihood of squirrel monkeys becoming detectably viremic (logistic regression, N = 7, Table 2, DF = 1, $\chi^2 = 1.0556$, P = 0.30). Given the small number of monkeys for which peak titer could be quantified, we did not analyze the association of dose with peak titer. DENV-2 was not detectable in any of the salivations from the mosquitoes that fed on squirrel monkeys at 3 and 4 dpi.

### Following infection with sylvatic DENV-2, higher NK cells on day 1 were associated with lower neutralizing antibody titers in both macaques and squirrel monkeys

We predicted, based on Hypothesis 2, that NK cells mobilized early in infection would show a negative relationship with peak virus titer and neutralizing antibody titer. To analyze variation in NK cells, we used the percent NK cells of the total number of peripheral blood mononuclear cells (PBMCs). After Bonferroni correction for multiple testing, in infected macaques, values for % NK cells at each dpi were significantly correlated with the following dpi, however values at D0 and D1 dpi were not significantly correlated. In infected squirrel monkeys, percent NK cells at earlier (1 or 2 dpi) and later (7 or 8 dpi) timepoints were significantly, positively correlated with each other (P = 0.005); earlier % NK cells were correlated with % NK cells prior to infection (P = 0.04), but later % NK cells were not (P = 0.21). Thus, we analyzed only early % NK cells (1 dpi for macaques or 1 and 2 dpi for squirrel monkeys, batched due to the cohort structure of the experiment).

Early % NK cells were not significantly different among control, low dose macaques (1 mosquito) and high dose macaques (>1 mosquito) macaques (ANOVA, DF = 2, F = 2.74, P = 0.11) (Fig. 2C), and Day 28 PRNT80 values were not significantly different between low and high dose macaques (t-test on $\log_{10}$ transformed data, DF = 7, t = 2.29, P = 0.056) (Table 1). Thus, we combined low and high dose macaques for analysis. Early % NK cells did not differ between control and high dose squirrel monkeys (ANOVA, DF = 1, F = 0.87, P = 0.37; note that all four control squirrel monkeys were used in comparisons with both DENV and ZIKV infected squirrel monkeys; Fig. 2C).

Due to the low level of replication of DENV-2, it was not possible to analyze peak titer for either macaques or squirrel monkeys. The effects of early % NK cells and host species, as well as the interaction between the two factors, on PRNT80 values in infected animals was tested using a linear model: the only significant effect was early % NK cells (ANOVA, DF = 1, F = 10.8, P = 0.005), and the relationship between early % NK cells and PRNT80 values was negative, as predicted (Fig. 2D).

### Sylvatic DENV-2 dynamics, transmission, and neutralizing antibody were similar between native and novel hosts

Based on Hypothesis 3, we predicted that transmission of DENV-2 would be less efficient from squirrel monkeys than cynomolgus macaques. Squirrel monkeys and macaques received similar doses of DENV-2 (Supplementary Text S1.1, S1.2, S1.3). When considering only infected animals, neither early % NK cells (t test, DF = 17, t = −0.39, P = 0.70) nor PRNT80 values (t test, DF = 17, t = 0.35, P = 0.73) differed between macaques and squirrel monkeys. Viremia was too low (Fig. 2A, B) and transmission was too infrequent (Fig. 3A, B) from both macaques and squirrel monkeys for statistical comparison but did not support the claim that transmission is lower from a novel host. Intriguingly, DENV-2 transmission from both macaques and squirrel monkeys was associated with low or undetectable viremia (Fig. 4A).

### Sylvatic ZIKV delivered by mosquito bite to both macaques and squirrel monkeys resulted in 100% seroconversion, but antibody responses differed between the species

ZIKV was detectable in the saliva of 22/25 intrathoracically inoculated mosquitoes that fed on macaques. All three macaques seroconverted (Table 2, Supplementary Dataset 5) with a minimum mean harmonic PRNT80 of 320 (treating the > 640 value as 640) (Fig. 2A). ZIKV was detectable in the saliva of 58/70 IT inoculated mosquitoes that fed on squirrel monkeys. There was a tight relationship between number of mosquitoes salivating ZIKV and total dose delivered (DF = 1, F = 12.6, P = 0.008, adjusted $R^2 = 0.56$, slope 0.14 (0.05; 0.22) in $\log_{10}$ units). All squirrel monkeys that received infectious mosquito bites seroconverted based on PRNT50 values

**Table 1 | Assignment of each cynomolgus macaque to low dose (1 infected mosquito) or high dose (10 infected mosquitoes) of DENV, or high dose (15 infected mosquitoes) of ZIKV, or control (10 uninfected mosquitoes) treatments and subsequent viremia, transmission to mosquitoes, and neutralizing antibody response**

| NHP ID | Sex | A priori Virus treatment | Post hoc Virus Treatment[a] | No. infected mosquitoes engorged | No. infected mosquitoes with detectable virus in saliva post passage[b] (total dose virus delivered ($\log_{10}$ pfu))[c] | Days on which virus was detectable in serum, day of peak titer (peak titer $\log_{10}$pfu/ ml) | Days on which virus was transmitted to mosquitoes (% infected)[d] | PRNT80 on Day -10 | 28 |
|---|---|---|---|---|---|---|---|---|---|
| NV289 | M | Control | Control | 8 | 0 | - | - | <20 | <20 |
| UG171 | F | Control | Control | 5 | 0 | - | - | <20 | <20 |
| NV259 | F | Control | Control | 9 | 0 | - | - | <20 | <20 |
| SB393 | M | Low Dose DENV | Low Dose DENV | 1 | 0 | - | - | <20 | 40 |
| FR469A | M | Low Dose DENV | Low Dose DENV | 1 | 1 | 4 (3.9) | - | <20 | 160 |
| UG253A[a] | M | Low Dose DENV | Control | 1 | 0 | - | - | <20 | <20 |
| BC407 | F | Low Dose DENV | Low Dose DENV | 1 | 1 | - | - | <20 | <20 |
| CP60 | F | Low Dose DENV | Low Dose DENV | 1 | 1 | - | - | <20 | 80 |
| BC116A | M | High Dose DENV | High Dose DENV | 7 | 6 | 4 (na[e]) | - | <20 | 40 |
| FR423A | M | High Dose DENV | High Dose DENV | 8 | 5 | 4, 7 (2.5) | 5 (60%) | <20 | 320 |
| SB395 | F | High Dose DENV | High Dose DENV | 8 | 3 | - | - | <20 | 1280 |
| FR840 | F | High Dose DENV | High Dose DENV | 9 | 4 | - | 8 (100%) | <20 | 320 |
| BC167 | F | High Dose DENV | High Dose DENV | 7 | 3 | - | 5 (20%) | <20 | 1280 |
| UGZ626 | F | High Dose ZIKV | High Dose ZIKV | 7 | 7 (4.41) | 2, 3, 4, 5 (5.0) | 2 (40%) 3 (54%) 4 (89%) 5 (70%) | <20 | >640 |
| EC644 | M | High Dose ZIKV | High Dose ZIKV | 11 | 9 (4.48) | 3, 4, 5 (5.3) | 3 (30%) 4 (100%) 5 (100%) | <20 | 160 |
| FR1221 | F | High Dose ZIKV | High Dose ZIKV | 7 | 5(4.42) | 3, 4, 5 (4.7) | 3 (9%) 4 (81%) 5 (100%) | <20 | 40 |

aSee text for justification of change from a priori to post hoc assignment.
bIncludes only mosquitoes that survived the two-day rest period post-feeding. Survival was generally very high; survival data is available in Supplementary Datasets 1 and 3.
cZIKV only; since viremia was never detected from unpassaged saliva for DENV-2, number of engorged mosquitoes was used as a proxy for dose delivered.
dConsidered positive if virus detected in bodies or legs, which were processed separately.
eNot applicable as viremia was detected only post-passage.

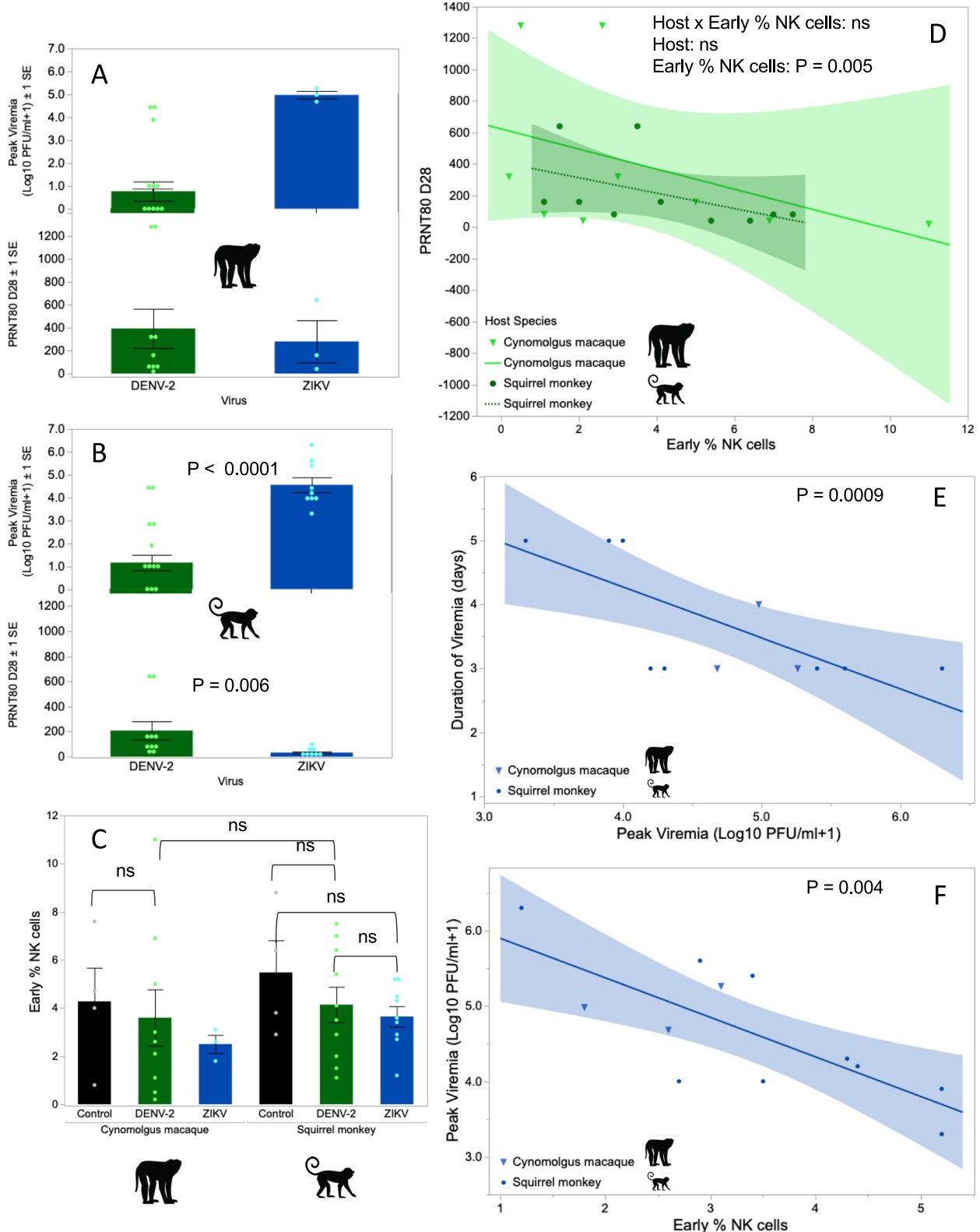

(Table 2, Supplementary Dataset 5), but 63% produced PRNT80 levels below the LOD, with a harmonic PRNT80 of 23.6 (Fig. 2B). The small sample size of ZIKV-infected macaques ($N = 3$) precluded most statistical analyses, but as described below, these data could sometimes be combined with squirrel monkey data to test the central hypotheses of the manuscript. As described in Supplementary Text S2, two squirrel monkeys infected with ZIKV had to be euthanized prior to the end of the experiment following the

recommendation of the head veterinarian on staff: 4683 due to cluster seizures and 4728 due to a worsening sore on the foot.

**Sylvatic ZIKV dynamics were similar in macaques and squirrel monkeys and peak viremia was negatively associated with duration of viremia**

All three ZIKV-infected macaques produced viremia that peaked between dpi 4 and 5 and lasted for 3 to 5 days (Table 1) and transmitted

**Fig. 2 | Effect of treatments on peak virus titer, 80% plaque reduction neutralization titer (PRNT80), and natural killer (NK) cells, and relationships among them, for dengue virus serotype 2 (DENV-2, in green) and Zika virus (ZIKV, in blue); values for individual animals shown as points. A** Top panel shows mean peak titer ± 1 SE of DENV-2 and ZIKV and bottom panel shows mean PRNT80 ± 1 SE, in cynomolgus macaques, $N = 13$ (10 DENV-2 and 3 ZIKV from a single independent experiment each) biologically independent animals in each panel. **B** Top panel shows higher mean peak titer ± 1 SE of ZIKV than DENV-2 (2-tailed $t$ test, DF = 17, $t = 7.18$, $P < 0.0001$), and bottom panel shows lower mean PRNT80 ± 1 SE against ZIKV than DENV-2 (Mann–Whitney $U$ test, DF = 1, chi-squared = 7.65, $P = 0.006$) in squirrel monkeys, $N = 19$ (10 DENV-2 and 9 ZIKV from a single independent experiment each) biologically independent animals in panel A and 18 (10 DENV-2 and 8 ZIKV from a single independent experiment each) biologically independent animals in panel (**B**), no adjustment was made for multiple tests. **C** Mean ± 1 SE of early % NK cells in macaques or squirrel monkeys do not differ between viruses or between infected and control animals (2-tailed ANOVA, DF = 4, $F = 0.44$, P = 0.78; note that ZIKV in cynomolgus macaques excluded from analysis due to small sample size), $N = 10^5$ total PBMCs per animal for 40 biologically

independent animals from a single independent experiment for each virus in each monkey species. **D** Early % NK cells are negatively associated with PRNT80 values for macaques (triangles) and squirrel monkeys (circles) infected with DENV-2, shaded band shows 95% confidence interval around the line of best fit (linear model, 2-tailed ANOVA, DF = 1, $F = 10.8$, $P = 0.005$), $N = 20$ biologically independent animals, with $10^5$ total PBMCs per animal used to estimate % NK cells. **E** negative relationship between peak viremia and duration of viremia for macaques (triangles) and squirrel monkeys (circles) infected with ZIKV, shaded band shows 95% confidence interval around the line of best fit (2-tailed ordinal logistic regression, DF = 1, chi-squared = 11.1, $P = 0.0009$, $R^2 = 0.17$), $N = 12$ biologically independent animals. **F** Negative relationship between early % NK cells and peak viremia for macaques (triangles) and squirrel monkeys (circles) infected with ZIKV, shaded band shows 95% confidence interval around the line of best fit (2-tailed linear regression, DF = 1, $F = 13.4$, $P = 0.004$, adjusted $R^2 = 0.53$), $N = 12$ biologically independent animals. No adjustment for multiple testing was made for any of the tests listed above. Images used under license from Shutterstock.com. Source data are provided as a Source Data file.

**Table 2 | Virus delivered to squirrel monkeys via cartons of 15 mosquitoes and subsequent viremia, transmission to mosquitoes, and neutralizing antibody response**

| NHP ID | Sex | Virus treatment | No. infected mosquitoes engorged | No. infected mosquitoes salivating virus[a] | Total dose of virus delivered (log$_{10}$ pfu) | Days on which virus was detectable in serum, day of peak titer, (peak titer log10 pfu/ml) | Days on which virus was transmitted to mosquitoes (% infected[b]) | PRNT80 on Day -7 | PRNT80 on Day 28 |
|---|---|---|---|---|---|---|---|---|---|
| 6314 | M | DENV | 3 | 2 | 0.3 | – | 1 (11%) 3 (8%) 7 (38%) | <20 | 640 |
| 6519 | M | DENV | 4 | 4 | 1.6 | – | – | <20 | 40 |
| 4516 | F | DENV | 5 | 4 | 3.2 | 1, 3, 7, 14 (1.9) | 1(100%) 5 (13%) | <20 | 640 |
| 5045 | F | DENV | 4 | 2 | 3.3 | – | – | <20 | 80 |
| 4872 | F | DENV | 3 | 3 | 1.6 | 3 (2.8), 14 | – | <20 | 160 |
| 6401 | M | DENV | 4 | 2 | 1.6 | – | 8 (22%) | <20 | 40 |
| 6363 | M | DENV | 5 | 2 | 0.3 | 2, 4, 14 (2.9) | 2 (44%) | <20 | 80 |
| 6552 | M | DENV | 4 | 3 | 2.3 | – | 2 (40%) | <20 | 80 |
| 6242 | F | DENV | 4 | 3 | 2.3 | – | 8 (57%) | <20 | 160 |
| 5910 | F | DENV | 6 | 5 | 2.3 | – | – | <20 | 160 |
| 6541 | M | Control | 2 | 0 | 0 | – | – | <20 | <20 |
| 6321 | M | Control | 4 | 0 | 0 | – | – | <20 | <20 |
| 6550 | M | ZIKV | 9 | 6 | 3.5 | 1, 3(4.0), 5 | 3 (35%) 5 (46%) | <20 | 80 |
| 6518 | M | ZIKV | 5 | 1 | 2.8 | 3 (3.3) | 1 (14%) 3 (14%) 5 (10%) | <20 | <20 |
| 6311 | M | ZIKV | 10 | 5 | 3.7 | 3 (3.9), 5 | 3 (50%) 5 (9%) | <20 | <20 |
| 5013 | F | ZIKV | 11 | 11 | 3.9 | 1, 3, 5 (4.0) | 3 (22%) 5 (71%) | <20 | <20 |
| 4806 | F | ZIKV | 4 | 4 | 2.9 | 3, 5 (5.6) | 5 (64%)[c] | <20 | <20 |
| 6347 | M | ZIKV | 6 | 6 | 3.2 | 4 (5.4) | 2 (17%) 4 (80%) | <20 | 40 |
| 6359 | M | ZIKV | 7 | 7 | 4.2 | 2, 4 (4.2) | 4 (60%) | <20 | <20 |
| 5730 | F | ZIKV | 4 | 4 | 3.1 | 2, 4 (4.3) | 4 (100%) | <20 | 40 |
| 4683 | F | ZIKV | 7 | 6 | 3.5 | 2, 4 (6.0), 8 | 2 (50%) 4 (25%) | <20 | na[d] |
| 4728 | F | ZIKV | 9 | 8 | 3.9 | 2, 4 (6.3) | 4 (50%) | <20 | na[d] |
| 5929 | F | Control | 8 | 0 | 0 | – | – | <20 | <20 |
| 5080 | F | Control | 10 | 0 | 0 | – | – | <20 | <20 |

[a]Includes only mosquitoes that survived the two-day rest period post-feeding. Survival was generally very high; survival data is available in Supplementary Datasets 2 and 4. When virus was only detectable after passage, the dose was arbitrarily estimated at 1 pfu per salivation.
[b]Considered positive if virus detected in bodies or legs which were processed separately.
[c]No mosquitoes that fed on Day 3 on this monkey survived incubation.
[d]Euthanized before Day 28.

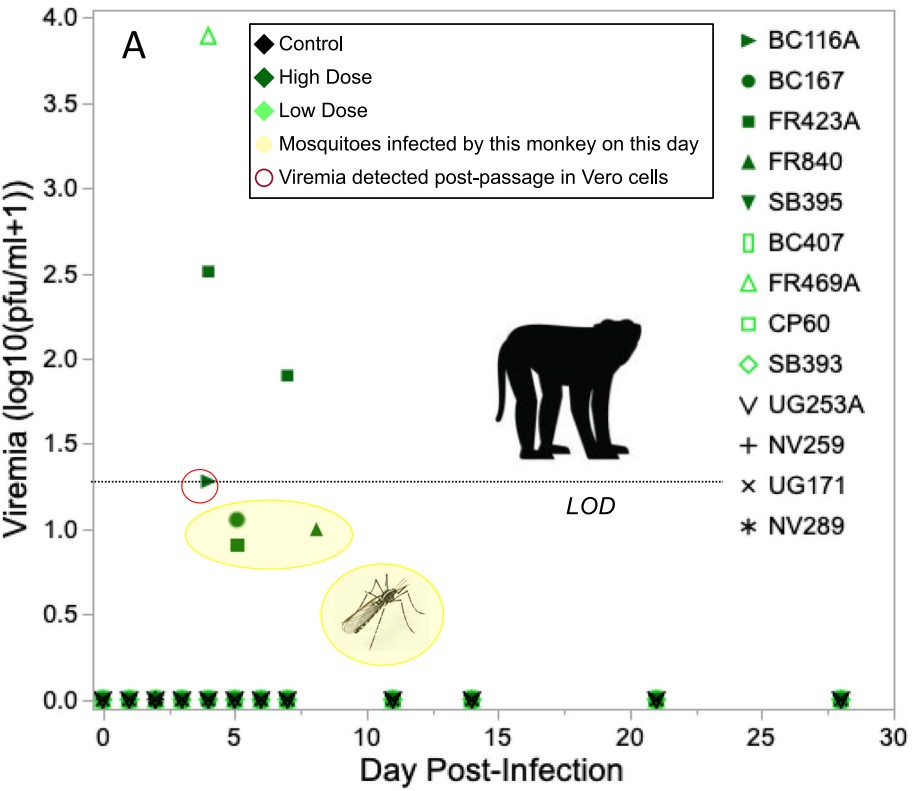

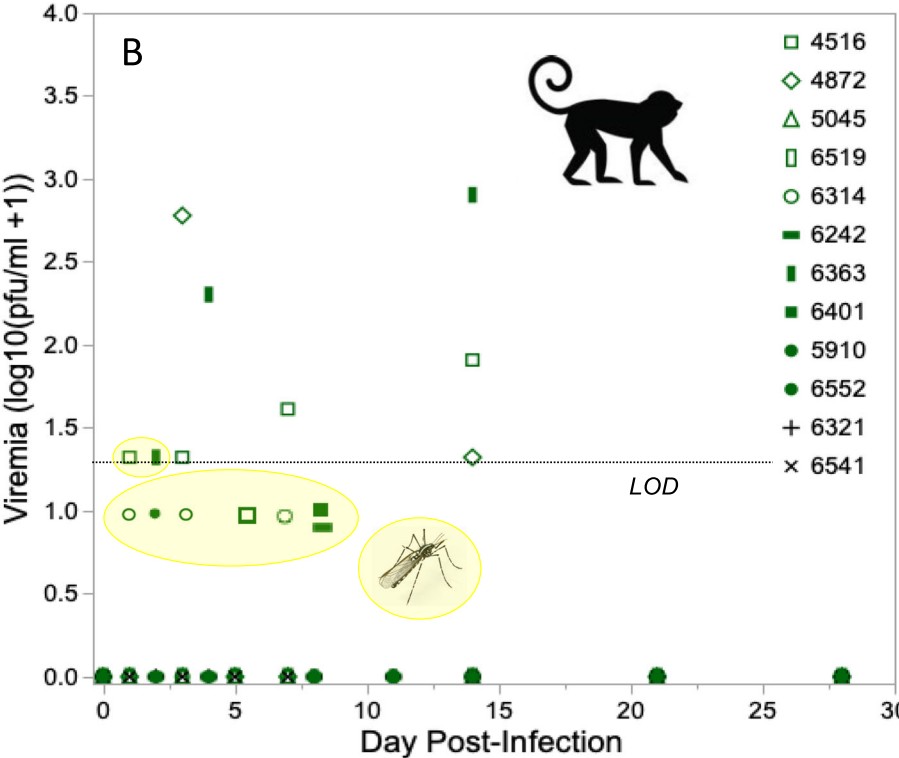

**Fig. 3 | Replication and transmission of sylvatic dengue virus serotype 2 (DENV-2) is similar in native (cynomolgus macaque) and novel (squirrel monkey) hosts.** Sylvatic dengue virus serotype 2 (DENV-2) replicates to low levels and transmits rarely to *Ae. albopictus* in infected: (**A**) cynomolgus macaques and (**B**) squirrel monkeys. Points for mosquito infections that occurred from monkeys without detectable viremia are placed at arbitrary positions below the limit of detection (LOD) for visualization only. Viremia was monitored by direct titration of serum as well as one passage of serum followed by titration; transmission was monitored by feeding cartons of 10-15 uninfected mosquitoes on each monkey; note that usually fewer than the full complement of mosquitoes engorged. Source data are provided as a Source Data file. Images used under license from Shutterstock.com.

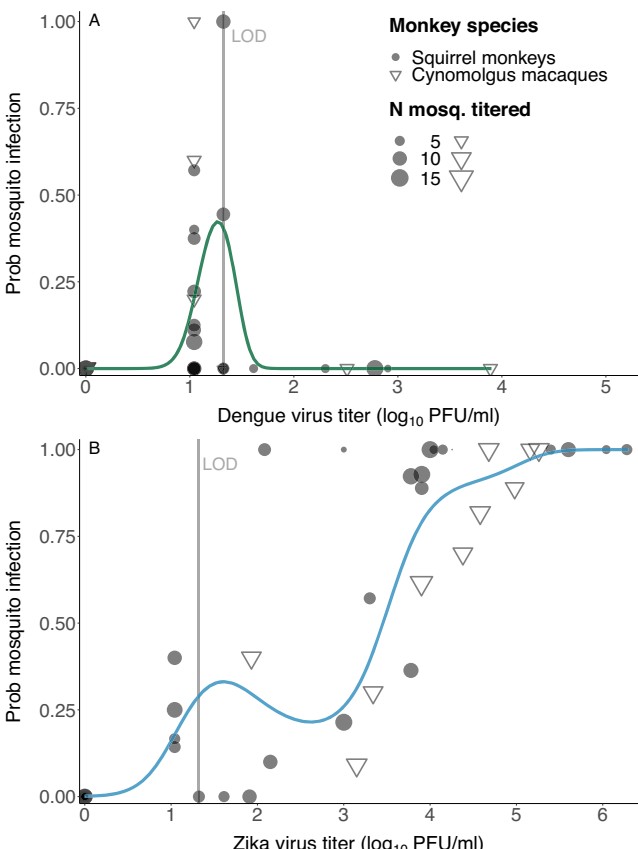

**Fig. 4 | Transmission of DENV-2 (in green) from non-human primates to mosquitoes occurred only when serum viremia was below or near the limit of detection, while ZIKV (in blue) transmission was positively associated with serum viremia.** Relationship between virus titer in non-human primate serum and transmission of (**A**) sylvatic dengue virus serotype 2 (DENV-2) or (**B**) sylvatic Zika virus (ZIKV) to *Ae. albopictus* (either body or leg infection). Curves fitted using a generalized additive model, allowing a variety of shapes, to unravel a possible pattern. Source data are provided as a Source Data file.

virus to a high proportion of mosquitoes during the window of viremia (Fig. 5A). All squirrel monkeys receiving bites from ZIKV-infected mosquitoes produced detectable viremia and infected mosquitoes on at least one dpi (Table 2, Fig. 5B, Fig. S3). ZIKV titer peaked between dpi 3 and 5 and viremia lasted between 3 to 7 days, although there is substantial uncertainty to these estimates given the sampling regimen.

For analyses of the magnitude and duration of ZIKV viremia in squirrel monkeys, we chose to exclude 4683, the animal euthanized at 13 dpi (Supplementary Text S2), as we did not know whether pre-existing health problems may ultimately have led to euthanasia and may have impacted viral replication or immune parameters; however, we note below when inclusion of this animal altered determination of the significance of comparisons. Contrary to the predictions of Hypothesis 1, the total dose delivered did not affect the peak titer of ZIKV in squirrel monkeys (DF = 1, F = 0.02, P = 0.88). Since every cynomolgus macaque received a higher virus dose of ZIKV than any squirrel monkey (see below), we did not analyze data from both species combined (Supplementary Text S1.3). In agreement with the predictions of Hypothesis 1, peak virus titer was negatively associated with the duration of infection for ZIKV, albeit the range of duration was quite limited (ordinal logistic regression, DF = 1, chi-squared = 7.4, P = 0.007, $R^2 = 0.17$). When these analyses were repeated with 4683 included, relationships were qualitatively similar, but the association between peak titer and duration of infection was no longer significant. Since both peak viremia and duration of viremia of ZIKV in macaques

fell within the range of these parameters in squirrel monkeys, we also conducted an analysis of combined values across species and found a similar, negative relationship (ordinal logistic regression, DF = 1, chi-squared = 11.1, P = 0.0009, $R^2 = 0.17$) (Fig. 2E).

### Following infection with sylvatic ZIKV, higher NK cells on day 1 were associated with lower peak virus titer in both macaques and squirrel monkeys

Values for early % NK cells in ZIKV-infected macaques all fell within the range of values for control macaques (Fig. 2C). Similarly, early % NK cells did not differ between ZIKV-infected and control squirrel monkeys (DF = 11, t = −1.72, P = 0.11; Fig. 2C). Moreover, % NK cells in infected squirrel monkeys at early and later timepoints were significantly, positively correlated with each other (P = 0.005) but not with % NK cells prior to infection (P > 0.27 for both comparisons). Consistent with predictions of Hypothesis 2, higher early NK cells were associated with lower peak virus titer in squirrel monkeys (DF = 1, F = 12.6, P = 0.009, adjusted $R^2 = 0.59$; this relationship was not significant when data from 4683 was included). Since the values for early % NK cells and peak viremia in macaques fell within the range of these parameters in squirrel monkeys, we also conducted an analysis of combined values across species and found a similar, negative relationship (DF = 1, F = 13.4, P = 0.004, adjusted $R^2 = 0.53$) (Fig. 2F). It was not possible to analyze the association between early NK cells and neutralizing antibody titer in squirrel monkeys, as approximately half of the PRNT80 values were <LOD.

### Sylvatic ZIKV transmission increased with increasing viremia, but novel host transmitted more efficiently than native hosts

In contrast to DENV-2, the likelihood of ZIKV transmission from both macaques and squirrel monkeys increased with virus titer (Fig. 4B, this analysis included 4683). Of mosquitoes that fed on dpi 3 and 4 on ZIKV-infected squirrel monkeys, one batch of mosquitoes died, but all other monkeys infected at least one mosquito that salivated detectable virus (Supplementary Dataset 6). The mean rate of virus dissemination to saliva from fed mosquitoes was 63% (±12%) (Supplementary Dataset 6).

The best dose-response fit to characterize the relationship between host viremia and disseminated ZIKV infection in mosquitoes (presence of virus in mosquito legs) was obtained using a logistic equation (Eq. S.1) and a betabinomial likelihood for both cynomolgus macaques and squirrel monkeys (Fig. 6A, Table S1). While the fits were not significantly different overall, the dose needed to infect 50% of mosquitoes ($ID_{50}$) was lower for squirrel monkeys (2.53, 95% CI (1.96;3.15) than macaques (3.84 (3.35;4.31) $\log_{10}$pfu/ml, Fig. 6A). When using data on mosquito body infection, the fitted relationships were significantly different between species (Table S2, Fig. S4A). This is because, as with DENV-2, among mosquitoes that fed on ZIKV-infected squirrel monkeys, a much higher proportion of infections could be detected in mosquito legs than bodies. However, these proportions were relatively similar among mosquitoes fed on ZIKV-infected macaques (Supplementary Dataset 6). The parameters of fitted relationships can be found in Table S3.

### Sylvatic DENV-2 and ZIKV dynamics differed dramatically from each other in mosquito vectors and monkey hosts

***Aedes albopictus***. Although *Ae. albopictus* were IT inoculated with the same dose of sylvatic DENV-2 and ZIKV, mean saliva titer was significantly lower in DENV-2 than ZIKV-infected mosquitoes (DENV 1.38 (1.24; 1.53), ZIKV 2.55 (2.36; 2.74) $\log_{10}$ pfu, P = 2.2e−4, Supplementary Text S1.2). In addition, the variability of saliva titers was also significantly lower in DENV-2 than ZIKV-infected mosquitoes (F = 12.7, P = 5e−4, Supplementary Text S1.2).

**Macaques.** Cynomolgus macaques received on average fewer bites from DENV-2 infected mosquitoes (mean 4.2, min−max (3−6) bites)

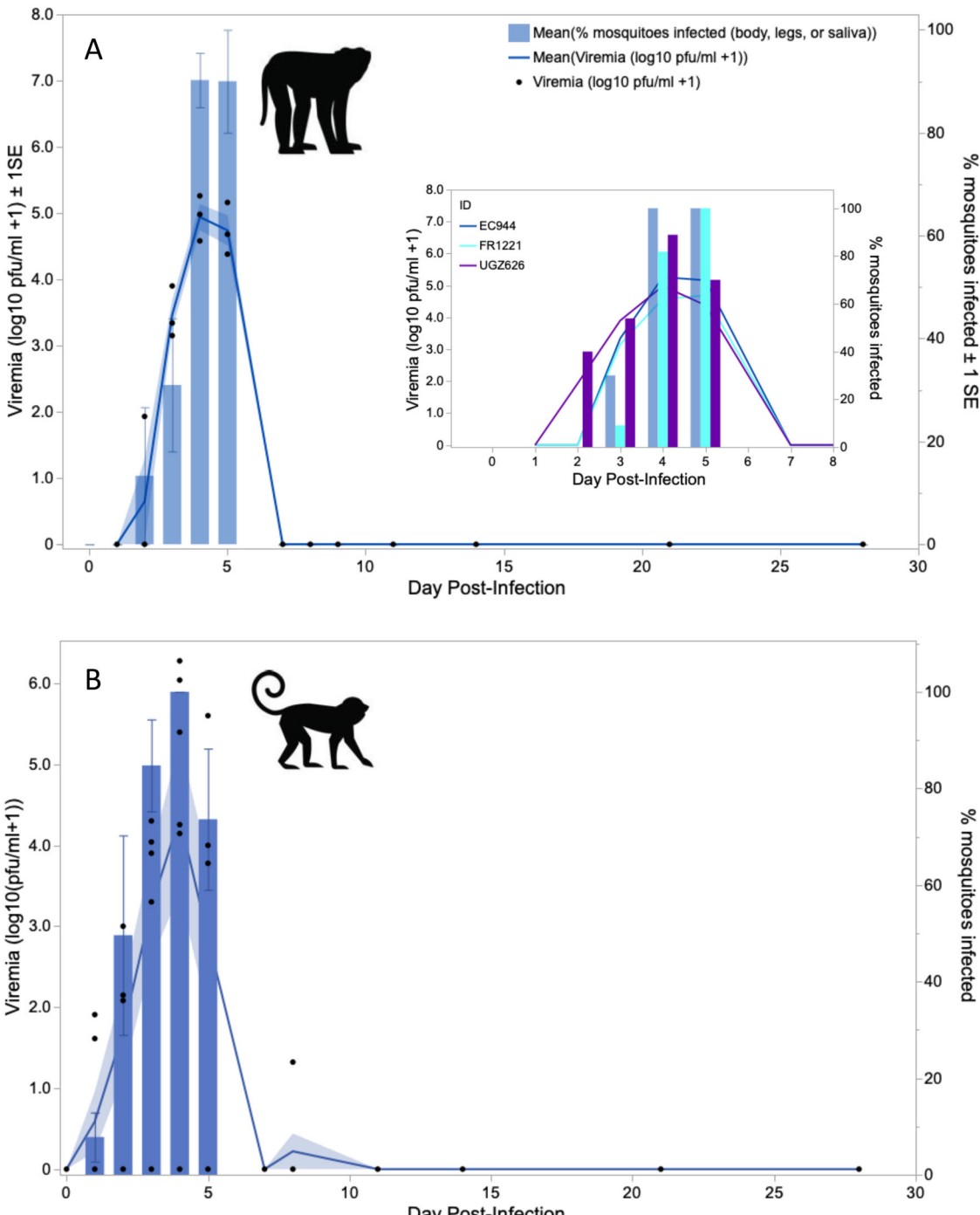

**Fig. 5 | Replication and transmission of sylvatic Zika virus (ZIKV) is similar in native (cynomolgus macaque) and novel (squirrel monkey) hosts.** ZIKV replicates robustly and transmits efficiently to *Ae. albopictus* in (**A**) cynomolgus macaques (*n* = 3 biologically independent animals) and (**B**) squirrel monkeys (*N* = 10 biologically independent animals) from a single independent experiment each. Control animals are not included in this figure. Blue lines indicate mean viremia, bands indicate ± 1 standard error (SE), and black dots indicate viremia for each individual monkey. Bars indicate mean % mosquitoes infected per monkey, and error bars indicate ± 1 standard error (SE). In panel A, the inset shows the dynamics of viremia and transmission during the window of viremia for each individual macaque. Viremia was monitored by direct titration of serum as well as one passage of serum followed by titration; transmission was monitored by feeding cartons of 15 uninfected mosquitoes on each monkey; note that usually fewer than 15 engorged. Source data are provided as a Source Data file. Images used under license from Shutterstock.com.

than ZIKV-infected mosquitoes (7.3 (5–10) bites), Supplementary Text S1.1) and the saliva titers of DENV-2-infected mosquitoes were significantly lower than those of ZIKV-infected mosquitoes, as shown above. Thus, macaques received lower doses of DENV-2 than ZIKV (DENV-2 mean 1.92, min-max (1.78-2.08) $\log_{10}$ pfu; ZIKV 4.44 (4.41-4.48), Supplementary Text S1.3). Although we did not conduct a formal analysis due to the small sample size of ZIKV-infected macaques, ZIKV reached peak titers that were, on average, ≥3 logs higher than DENV-2 in macaques (Fig. 2A), yet early NK cells and neutralizing antibody titers were quite similar, with all values for ZIKV-infected macaques falling within the range of DENV-2 infected macaques (Fig. 2A, C).

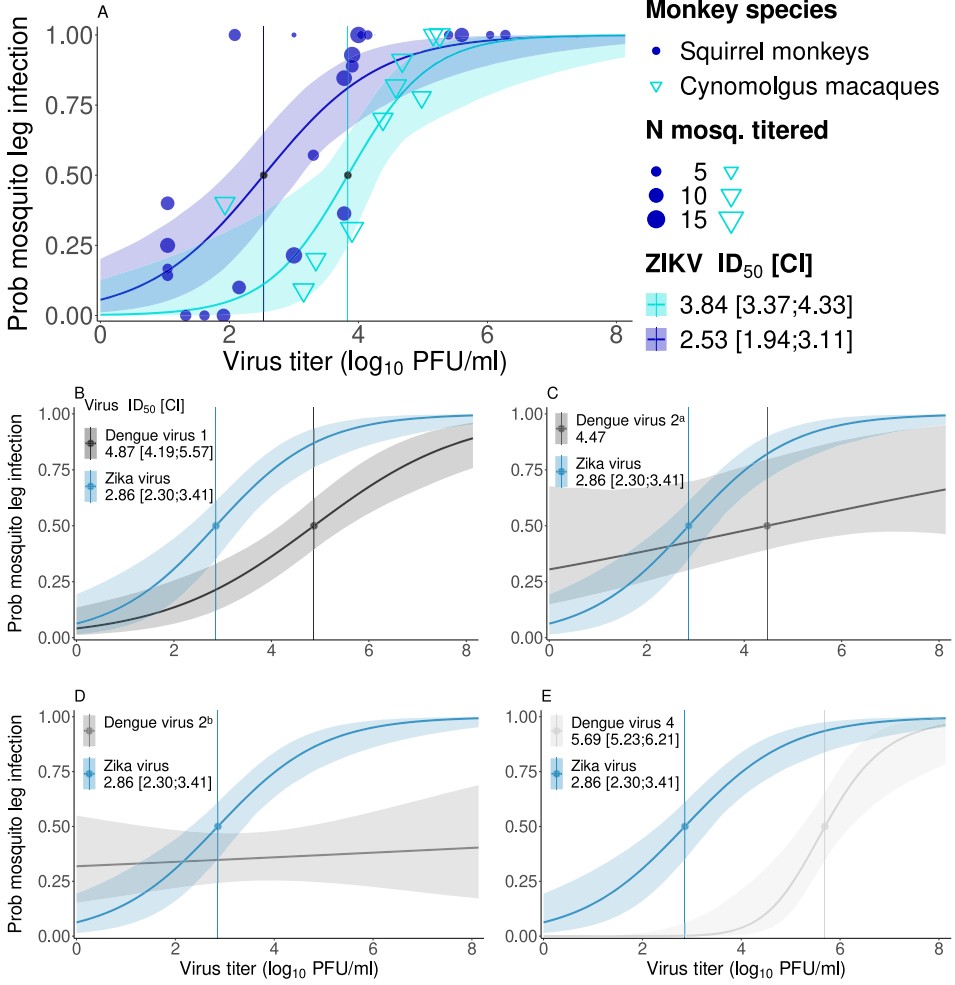

**Fig. 6 | Novel hosts transmitted ZIKV to mosquitoes more efficiently than native hosts. A** Relationship between virus titer in serum and transmission of sylvatic Zika virus (ZIKV) to legs of *Ae. albopictus* from cynomolgus macaques (light blue) or squirrel monkeys (dark blue). Points show raw data, with point size proportional to the number of mosquitoes tested in the batch (one day in one monkey). **B**–**E** Relationships between virus titer in serum and transmission of ZIKV to legs of *Ae. albopictus* from both non-human primate species combined (blue curve, repeated in **B**–**E**) or transmission to legs of *Ae. aegypti* from humans of (**B**) Dengue virus (DENV) serotype 1, (**C, D**) DENV serotype 2, (**E**) DENV serotype 4. DENV curves were fitted to data from Duong et al.[51] for sub-panels (**B, C, E**), and to data from

Long et al. 2019[52] for sub-panel (**D**). Infectious dose 50 (ID$_{50}$) and confidence interval (CI) provided for each designated virus and mosquito species in log$_{10}$ pfu/ml. Model selection was conducted with three competing functional forms, all with a sigmoidal shape. Curves represent the best fit, and shaded areas correspond to 95% confidence intervals of the fitting process. ª CI of ID$_{50}$ not provided as it was too wide to be insightful. ᵇ ID$_{50}$ and CI not provided as they were not insightful. Note that data used for fitting of this DENV-2 curve was from mosquito heads rather than legs, both are markers of a disseminated infection. Source data are provided as a Source Data file.

**Squirrel monkeys.** Squirrel monkeys received on average fewer bites from DENV-2-infected mosquitoes (3.4 (2-6)) than ZIKV-infected mosquitoes (6.0 (1-11)) and the saliva titers of DENV-2-infected mosquitoes were significantly lower than those of ZIKV-infected mosquitoes (DENV-2 1.38 (1.24;1.53); ZIKV 2.55 (2.36;2.74) log$_{10}$ pfu. Thus, squirrel monkeys received lower doses of DENV-2 than ZIKV (DENV-2 2.65 (1.60-3.37) log$_{10}$ pfu; ZIKV 3.68 (2.81-4.20), Supplementary Text S1.3. Even though ZIKV peak titers in squirrel monkeys were > 3 logs higher than DENV-2 titers (*t* test, DF = 17, *t* = 7.18, *P* < 0.0001; Fig. 2B, Tables 1 and 2), PRNT80 values (with values < LOD set to 19) were significantly lower in ZIKV-infected than DENV-2-infected squirrel monkeys (Mann–Whitney *U* test, DF = 1, chi-squared = 7.65, *P* = 0.006) (Fig. 2B). Early % NK cells, in contrast, were similar between the two viruses (*t* test, DF = 17, *t* = 0.56, *P* = 0.58) (Fig. 2C). In a reduced dataset with similar range of doses for DENV and ZIKV ((2.3–3.3) log$_{10}$ pfu; *N* = 5 DENV-2 and 4 ZIKV), peak viremia was still significantly higher in ZIKV- than DENV-2-infected squirrel monkeys (DF = 1, *F* = 37.83, *P* = 0.0005, adjusted *R*$^2$ = 0.82).

**Across hosts.** Patterns of DENV-2 and ZIKV replication summed across hosts (Fig. S5) were dramatically different. DENV-2 sustained low but prolonged viremia while ZIKV peaked quickly and was quickly cleared. The transmission also differed, as highlighted when fitting generalized additive models to each dataset (Fig. 4, Supplementary Text S3). The smooth term associated with virus titer was non-significant for DENV (*P* = 0.118) and significant for ZIKV (*p* < 2e−16). The deviance explained by both models was 67.4% and 85.2%, respectively.

**Relationship between viremia titer and transmission is similar between ZIKV in NHPs and DENV in humans**

Using a common fit for both species, we compared the relationship for disseminated infection to the ones obtained using data generated by Duong et al.[51] and Long et al.[52] in which DENV viremia (serotypes 1, 2, and 4) in infected humans and transmission to *Ae. aegypti* mosquitoes (measured as infection of mosquito legs for Duong et al., mosquito heads for Long et al.) were quantified (Fig.6B–E). The ID$_{50}$ for ZIKV transmission from NHPs to *Ae. albopictus* was substantially lower than

for DENV transmission from humans to *Ae. aegypti* (Fig. 6B–E). We conducted a similar comparison to data on DENV transmission (all four serotypes) from infected humans to *Ae. aegypti* generated by Nguyen et al.[53] and Long et al.[52], focusing on transmission measured as infection of mosquito bodies. In this case, there was much overlap between ZIKV transmission from NHPs and DENV transmission from humans (Fig. S4B). Fitted relationships for DENV with the corresponding data are shown in Fig. S6.

## Discussion

This study investigated whether the replication and clearance of arboviruses are locked into an immunologically-mediated trade-off, whether NK cells may be associated with this trade-off, and whether the nature of the trade-off differs between native and novel NHP hosts. If higher virus replication does drive faster clearance, we predicted that increasing viral dose would result in higher peak virus titer, that transmission would increase with virus titer, and that peak titer would be inversely related to duration of viremia. To test these predictions, we exposed cynomolgus macaques and squirrel monkeys to the bites of mosquitoes infected with either sylvatic DENV-2 or sylvatic ZIKV and monitored viremia as well as forward transmission to uninfected mosquitoes. Virus dose was estimated by summing the amount of virus in forced salivations from each mosquito that had fed upon a particular animal. While forced salivations likely underestimate the absolute amount of virus delivered[54,55], they should reflect the relative dose of virus delivered. Our previous study of mosquito delivery of ZIKV to mice found a highly significant increase in neutralizing antibody titer with increasing virus dose estimated via forced salivations[56].

Eight macaques and ten squirrel monkeys bitten by sylvatic DENV-2-infected mosquitoes seroconverted and produced robust neutralizing antibody titers. However, viremia was detected via cell culture assay in only three macaques and three squirrel monkeys, and only on one or a few days per animal. Intriguingly, nine animals in total transmitted DENV-2 to mosquitoes, but transmission usually occurred when the virus was not detected in the blood. This "decoupling" of transmission from DENV viremia has also been observed in studies in which *Ae. aegypti* were fed on viremic humans: Lambrechts et al.[57] found that, at the same level of viremia, humans infected with DENV-2 transmitted to more mosquitoes in earlier days after onset of illness than later days, and Duong et al.[51] found that people infected with DENV transmitted more efficiently to mosquitoes when they were asymptomatic and presymptomatic than when they were symptomatic, independent of viremia level. A number of host or viral products could be responsible for curbing transmission, such as blood biochemistry, immune factors, or viral proteins. Our data suggest that whatever is responsible is rather labile, as, in some monkeys (for example, macaque FR423A and squirrel monkey 4516), viremia could be detected from blood on the days before and after the monkey transmitted virus to mosquitoes but not on the day of transmission itself. With respect to our central hypothesis, it was not possible to analyze peak titer in either species due to the small number of monkeys with quantifiable viremia; we did find that a higher number of infectious mosquito bites increased likelihood of infection of macaques but not squirrel monkeys.

In contrast to DENV-2, sylvatic ZIKV reached high titers in both host species and transmission to mosquitoes was positively related to virus titer. In infected mosquitoes, ZIKV disseminated efficiently to the saliva. Previous studies have shown similar levels of infection, dissemination and salivation of ZIKV strain 41525 in multiple strains of both *Ae. albopictus* and *Ae. aegypti* using multiple infection paradigms[58–60], suggesting that our results for this ZIKV strain are generalizable across domestic and peridomestic *Aedes* vectors. Although both host species supported high levels of viremia and transmission of ZIKV, cynomolgus macaques raised robust neutralizing antibody titers that were equivalent to those produced against DENV-2, while squirrel monkeys raised neutralizing antibody titers that

were significantly lower than those of DENV-2, and often below the LOD when a stringent cutoff (PRNT80) was used. Similarly, studies on Ross River virus (reviewed in[61]) showed that experimentally infected bats produced no detectable viremia but nonetheless infected mosquitoes, while corellas (*Cacatoes correla*) became viremic but did not seroconvert. In combination, these findings highlight the complexity of using serological data to discern the role of species in virus transmission cycles[62,63].

A trade-off between magnitude and duration of infection was evident both within and between viruses. For ZIKV infections, maximum virus titer was associated with a shorter duration of infection. Comparing between viruses, ZIKV titers were significantly higher than DENV-2 titers in both cynomolgus macaques and squirrel monkeys, but the monkeys infected with ZIKV were viremic for only three to seven days while the monkeys with detectable DENV-2 viremia were viremic over twelve to fourteen days. Because higher titers of ZIKV were expectorated into mosquito saliva, it is not entirely possible to determine whether this difference in dynamics is due to the virus per se or to the higher dose of ZIKV delivered, although analysis of a subset of the monkeys in which dose was equivalent still found higher peak titers of ZIKV. However, since the ZIKV produced higher virus levels in saliva than DENV after equivalent amounts of virus were IT injected into mosquitoes, and since these results are consistent with a previous study by Chaves et al.[64] in which *Ae. aegypti* were infected with both viruses via membrane feeding, we consider the high dose of ZIKV intrinsic to the phenotype of this strain. Moreover, the low levels of DENV-2 replication in this study are consistent with results from our previous study of the replication of an African sylvatic DENV-2 strain in African green monkeys (*Chlorocebus sabaeus*) following needle delivery of a high dose ($10^5$ pfu) of the virus[24]. Interestingly, the effect of virus titer on transmission to mosquitoes was different for DENV-2 and ZIKV, with DENV-2 transmission occurring when viremia was low or just above the LOD, while ZIKV transmission was positively associated with titer. Thus, it seems that sylvatic DENV-2 follows a "tortoise" strategy in squirrel monkeys, while sylvatic ZIKV follows a "hare" strategy[32].

A previous mathematical model of the replication-clearance trade-off in DENV utilized NK cells as a proxy for the innate immune response to the virus[36]. We predicted that if NK cells are coupled to the immune responses shaping this trade-off, then higher mobilization of NK cells early in infection should suppress peak virus titer, which should in turn lower neutralizing antibody titers. For sylvatic DENV-2, peak titer could not be analyzed, but as the percentage NK cells out of total PBMCs on early dpi increased, PRNT80 values measured at 28 dpi decreased in both species, consistent with our prediction. The relationship of peak ZIKV viremia with PRNT80 values could not be analyzed, but peak ZIKV titers were negatively associated with higher early NK cells in both host species. Thus, there is substantial support that NK cells may play a direct role in shaping the replication of these viruses[65] or may be linked to another pathway that enacts this function.

Third, we predicted that transmission from novel hosts would be low relative to native hosts, reflecting the co-adaptation of viruses with their native hosts. Infection of both cynomolgus macaques and squirrel monkeys with sylvatic DENV-2 via mosquito bite resulted in muted virus replication. Previous studies utilizing needle delivery of human-endemic DENV have reported peak virus titers about three orders of magnitude lower in cynomolgus macaques than neotropical primates[31]. *Prima facie*, these latter data would suggest that transmission is actually more efficient from neotropical monkeys than Old World monkeys. However, our startling finding that sylvatic DENV transmission to mosquitoes occurred predominantly at undetectable or barely detectable viremia indicates a more complex relationship between viremia and transmission. Nonetheless, transmission of sylvatic DENV-2 from both macaques and squirrel monkeys was infrequent. This paucity of transmission could be the byproduct of a

mismatch in the genetic interaction between sylvatic DENV-2 and *Ae. albopictus* Galveston[66] or could be an artifact of the passage history of this virus strain. Moreover, it is possible that sylvatic mosquito species, such as *Aedes niveus* species, are more susceptible to sylvatic DENV. Further study of transmission to sylvatic African, Asian and American mosquitoes would certainly be welcome; but few of these species are maintained in colonies. Alternatively, the low level of transmission could suggest that cynomolgus macaques are not actually a reservoir host for this virus. The paradigm that Asian NHPs are the reservoir of sylvatic DENV, rather than an amplifying host or a dead-end host[67], rests on scanty evidence, including serosurveys and isolation of sylvatic DENV from sentinel cynomolgus macaques[37–39]. On the other hand, the lack of pathogenesis of sylvatic DENV-2 in macaques, the long duration of viremia, and indeed the low magnitude of viremia itself may be evidence that macaques are reservoir hosts[68,69]. Going forward, we will attempt to parameterize mathematical models with these data to assess whether or not cynomolgus macaques could be expected to sustain transmission of sylvatic DENV.

We found that ZIKV replicated to approximately a log higher titer in squirrel monkeys than macaques and, controlling for titer, transmitted to mosquitoes more efficiently from squirrel monkeys than macaques. Similarly, a Puerto Rican strain of ZIKV delivered by needle replicated to peak titers that were 2–3 orders of magnitude higher in neotropical marmosets and tamarins compared to rhesus and cynomolgus macaques[70]. Dudley et al.[33] found that, compared to needle delivery, mosquito bite delivery of a Puerto Rican strain of ZIKV to rhesus macaques delayed the onset of viremia and delayed peak of viremia from day 3 to day 5 pi, but for both methods of delivery, almost none of the *Ae. aegypti* fed upon these monkeys became infected. We reported similarly low transmission from cynomolgus macaques infected with a Cambodian strain of ZIKV via needle delivery and fed upon by *Ae. aegypti*[71]. Thus, opposite our initial hypothesis, evidence to date suggests that neotropical primates support similar or higher levels of DENV and ZIKV transmission than their Old World counterparts. Moreover, similar to previous findings by Carrington et al.[72], we detected little difference in replication dynamics of DENV-2 and ZIKV after delivery by mosquito bite and studies in which the viruses were delivered by needle.

Finally, our data offer insights into two other central questions: why has DENV never spilled back into a sylvatic cycle in the Americas and what is the likelihood that ZIKV will do so? The rarity of transmission of sylvatic DENV-2 from squirrel monkeys to *Aedes albopictus* suggests that the virus cannot gain a foothold in neotropical NHPs and sylvatic mosquitoes, albeit testing more strains of mosquitoes and monkeys, as well as modeling, are still needed. In contrast, the viremia-transmission curves of sylvatic ZIKV in squirrel monkeys were similar to those derived from studies of DENV-1-4 in humans[51–53]. While additional modeling will be needed to draw inferences about the epidemic potential of *Ae. albopictus* for ZIKV in neotropical monkeys, as was done by Lequime et al. for a human population[73], our empirical data suggest that the establishment of a neotropical, enzootic cycle is substantially more likely for ZIKV than DENV.

## Methods

### Ethics statement
Our study complies with all relevant ethical guidelines and all community standards for containment of infected arthropod vectors; all procedures conducted on non-human primates were approved via UTMB Institutional Animal Care and Use Committee (IACUC) protocol 1912100, approved on December 1, 2019.

### Viruses and cell lines
African green monkey kidney cells (Vero, catalog number CCL-81) and larval *Ae. albopictus* cells (C6/36, catalog number CRL-1660) were purchased from the American Type Culture Collection (ATCC,

Bethesda, MD, USA). Vero cells were maintained in Dulbecco's Modified Eagle's Medium (DMEM, ThermoFisher Scientific, Waltham, MA, USA) supplemented with 5% (v/v) heat-inactivated fetal bovine serum (FBS, Atlanta Biologicals, Flowery Branch, GA, USA), 1% (v/v) Penicillin–Streptomycin (P/S, ThermoFisher Scientific, Waltham, MA, USA; 100 U/mL and 100 µg/mL, respectively) in a humidified 37 °C incubator with 5% $CO_2$. C6/36 cells were maintained in DMEM supplemented with 10% (v/v) heat-inactivated FBS, 10% (v/v) tryptose phosphate broth (TBP, Sigma-Aldrich, St. Louis, MO, USA), 1% (v/v) P/S in a humidified 28 °C incubator with 5% $CO_2$.

The following viruses with the following passage histories were utilized to infect monkeys: sylvatic DENV-2 strain P8-1407 (suckling mouse (SM) 3 passages, C6/36 5 passages) and sylvatic ZIKV strain DakAR 41525 (AP-61 1 passage, C6/36 2 passages, Vero 6 passages). The DENV-2 stock was 3 C6/36 passages removed from a stock sequenced by Vasilakis et al.[74] (Genbank ID KU95559.1), and the ZIKV stock was 3 C6/36 passages removed from the stock sequenced by Ladner et al.[75] (Genbank ID EF105379.1). A diagnostic region of the envelope gene of each working stock of virus (nucleotides 1435–1744 for DENV-2 and 2163–2498 for ZIKV) were sequenced and showed no changes relative to the reference sequence. Pre-study blood samples were screened via PRNT assays against DENV-1 Hawaii, DENV-2 NGC (macaques) or DENV-2 16681 (squirrel monkeys), DENV-3 H87, DENV-4 Dominica, ZIKV PRVABC-59, and YFV 17D; all were negative (Supplementary Dataset 5; post-infection PRNTs were screened against DENV-2 NGC (macaques), DENV-2 16681 (squirrel monkeys) or ZIKV PRVABC-59 (both species), as appropriate). The switch between DENV-2 strains was necessitated by a freezer failure, but in our experience, should have little impact on PRNT values.

### Virus quantification from cell culture supernatants, NHP sera and mosquitoes
Virus stocks and NHP sera were titered in Vero cells while mosquito samples were titered in C6/36 cells. All NHP sera were titered starting at a 1:10 dilution and were also passaged once in Vero cells for 5–7 days as follows: generally, 500 µL of 1:10 diluted NHP serum in Vero cell media was used to infect one well of confluent Vero cells in 6-well plates, plates were gently rocked for two hours before adding media to a total volume of 3 mL and then plates were incubated for 4-6 days, cell supernatants were clarified by centrifugation, stabilized in 1X SPG (final concentration: 218 mM sucrose, 6 mM L-glutamic acid, 3.8 mM monobasic potassium phosphate, and 7.2 mM dibasic potassium phosphate, pH 7.2), and stored at −80˚C. For a relatively small number of serum samples with volumes <50 µL, dilutions of 1:20, 1:40 and 1:100 were used as necessary. Resulting cell supernatants were titered in Vero cells. Fifty µl of saliva samples were titered and were also passaged once in one well of a 96-well plate of confluent C6/36 cells for 5-7 days, and the resulting cell supernatants were titered in C6/36 cells.

Viral quantification of cell-free viral stocks for infection and PRNT assays were conducted using standard infectious assay[71]. Briefly, viral stocks underwent 10-fold serial dilutions using dilution media (DMEM, 2% FBS, and 1% P/S) in sterile 96-well tissue culture plates. Dilutions were added to 85-95% confluent monolayers of either Vero or C6/36 cells, as specified, in 12 or 24 well tissue culture plates. Dilutions were allowed to adsorb onto monolayers for 1 hour in a humidified 37 °C (Vero) or 28 °C (C6/36) incubator with 5% $CO_2$ and were rocked every 15 minutes to prevent drying of the monolayers. Following adsorption, monolayers were overlayed using a solution of DMEM containing 3% FBS, 1% Pen–Strep, 1.25 µL/mL amphotericin B, and 0.8% weight/vol methylcellulose. The overlayed plates were incubated for 5 days in a humidified 37 °C incubator with 5% CO2; then, each well was washed twice with PBS and fixed for a minimum of 30 min in an ice-cold solution of methanol:acetone, 1:1, vol/vol. The fixative was removed, and the plates were air dried. Following complete air drying, the plates were washed with PBS and then blocked with 3% FBS in PBS, followed

by an overnight incubation with mouse hyperimmune serum against DENV-2 NGC (MIAF T-35432, generated 12/10/2014) or mouse hyperimmune serum against ZIKV strain MR-766 (MIAF T-36846 generated 07/19/2017) or pan-flavivirus monoclonal antibody 4G2. All three antibodies were used at a 1:2000 dilution; MIAF T-35432 and MIAF T-36846 were obtained from the World Reference Center on Emerging Viruses and Arboviruses (WRCEVA) at University of Texas Medical Branch while 4G2 was obtained from VWR (Radnor, PA, catalog number 76005-718). The plates were then washed with PBS, followed by incubation with a goat anti-mouse secondary antibody conjugated to horseradish peroxidase (KPL, Gaithersburg, MD, USA, catalog number 5220-0341) diluted 1:2000 in blocking solution. The plates were washed with PBS, after which an aminoethylcarbazole solution (Enzo Diagnostics, Farmingdale, NY, USA) prepared according to the manufacturer's protocol was added, and the plates were incubated in the dark. Development was halted by washing in tap water, and the plates were allowed to air dry at room temperature before scoring. Virus quantification from monkey sera and mosquito samples was conducted using standard infectious assays, which are extremely similar to those described above save that titration was conducted in 24-well plates, plates were fixed in 90% methanol at room temperature, incubated with primary antibodies for one hour, and developed using TrueBlue (SeraCare, Milford, MA, USA).

## Mosquito strains, maintenance, infection, and forced salivation

*Aedes albopictus* were field collected in Galveston Texas in the summer of 2018 and utilized to establish a continuous colony. All mosquitoes utilized over the course of these experiments were derived from this original colony. For infection of cynomolgus macaques with DENV-2 and control, *Ae. albopictus* Galveston F12 were used, while for infection of cynomolgus macaques with ZIKV and squirrel monkeys with both DENV-2 and ZIKV, *Ae. albopictus* Galveston F14 were used. All mosquitoes were maintained and/or utilized in Biosafety Level 2 facilities (Arthropod or Animal Containment) following standard procedures[76]. After hatching, larvae were maintained in water in plastic containers, and then transferred after eclosion into 0.5 L cardboard cups overlayed with mesh across the lids, and cartons were held in Tupperware containers within insect incubators set to $28 \pm 1\,°C$ with $80 \pm 10\%$ RH and a 16:8 light:dark cycle. Mosquitoes were provided *ad libitum* access 10% sucrose-soaked cotton rounds, save that one day prior to feeding on NHPs these were replaced with water-soaked rounds.

For mosquitoes used to infect monkeys, females at 4 days post-eclosion were IT inoculated with between 200 to 500 pfu of either DENV or ZIKV in a volume of $0.2\,\mu L$ via a Nanoject II (Drummond Scientific, Broomall, PA, USA) and held in cartons as described. Ten days later, mosquitoes were cold-anesthetized and sorted into cartons based on treatment and individual NHP. Mosquitoes were then allowed to feed on monkeys as described below, engorged mosquitoes were separated and housed for 2-3 days. In high-dose treatments, engorgement rates per carton were always <100%; in low-dose treatments, if the single mosquito did not feed, which occurred twice, the feeding was repeated the following day with a new, IT-infected mosquito. To harvest saliva, mosquitoes were cold anesthetized and immobilized on mineral oil, after which legs and wings were removed and the proboscis was inserted into sterile $10\,\mu L$ pipette tips containing $10\,\mu L$ of FBS for 30 minutes. After this, the saliva within the FBS was ejected into a microfuge tube containing $100\,\mu L$ of DMEM containing 2% FBS, 1% P/S and $2.5\,\mu g/mL$ of amphotericin B (hereafter referred to as mosquito collection media, MCM).

For mosquitoes used to detect transmission, following feeding, all mosquitoes were cold-anesthetized and engorged mosquitoes were separated and housed for 14 days. They were then cold anesthetized on ice and the legs of individual mosquitoes were removed; bodies and legs were placed in separate microfuge tubes containing sterilized steel ball bearings and $500\,\mu L$ of MCM. In mosquitoes fed

on squirrel monkeys at dpi 3 and 4, saliva was also collected as described above.

## Non-human primate origins, maintenance, and monitoring

All procedures conducted on non-human primates were approved via UTMB Institutional Animal Care and Use Committee (IACUC) protocol 1912100, approved on December 1, 2019. Mauritius-origin adolescent *Macaca fascicularis* of both sexes (between 2.4 to 5.4 years of age) weighing between 2.75-5.4 kg were purchased from Worldwide Primates, Inc (Miami, FL, USA). All animals were negative by serology for macacine alphaherpesvirus 1, simian immunodeficiency virus (SIV), simian type D retrovirus (SRV), simian T-lymphotropic virus 1 (STLV-1). Following a minimum 14-day quarantine, all animals were surgically implanted with DST micro-T temperature loggers (Star-Oddi, Garðabær, Iceland) set to record temperature every 15 minutes. Macaques were single housed in open metal caging that allowed for visual but not physical contact with other animals in the room. Standard primate chow was provided twice daily, with enrichment in the form of fruits and vegetables added once daily. All animals received a minimum of twice daily health checks.

*Saimiri boliviensis boliviensis* (Black capped squirrel monkeys) of both sexes over an adolescent to adult age range (between 4 to 14 years of age) were purchased from the MD Anderson Center (Bastrop, TX, USA). Following a minimum 14-day quarantine, all animals were surgically implanted with DST micro-T temperature loggers (Star-Oddi, Garðabær, Iceland) set to record temperature every 15 minutes. Squirrel monkeys were pair housed in open metal caging that allowed for visual but not physical contact with animals in other cages. Standard primate chow was provided twice daily, with enrichment in the form of fruits and vegetables added once daily. All animals received a minimum of twice daily health checks.

## Non-human primate infection, blood sampling and mosquito feeding

All experimental procedures requiring any manipulation of the monkeys were conducted following an overnight fast and under anesthesia, via intramuscular injection of 5-20 (cynomolgus macaques) or 10-15 (squirrel monkey) mg/kg ketamine. Once anesthetized, animals were removed from caging, and physical examination and weights were taken. To maintain core body temperature in squirrel monkeys following anesthesia, all procedures took place atop a preheated heating blanket. To infect monkeys, a 0.5 L cardboard cup containing either 1, 10 or 15 (Fig. 1) sucrose-starved *Ae. albopictus* Galveston previously injected with DENV-2 or ZIKV or 10 or 15 uninjected control *Ae. albopictus* was placed on the anesthetized animal's ear and the mosquitoes were allowed to feed for 7-10 minutes. On subsequent time points following infection, cartons of 10 (DENV-2-infected and control cynomolgus macaques) or 15 (all other experiment arms) sucrose-starved naïve *Aedes albopictus* were allowed to feed on the ear of the anesthetized animals.

Concurrent with mosquito feeding, blood was collected via venipuncture of the femoral vein. To measure viremia and PRNTs, blood was aliquoted into serum separator tubes (SSTs, BD Biosciences) and centrifuged at 2000g for 5 min to pellet cells and clarified serum was aliquoted into fresh tubes. The others were retained for viremia and PRNT analysis. For the quantification of NK cells, whole blood from venipuncture was added to lithium heparin microtainer tubes (BD Biosciences) and gently inverted.

## Natural killer cell quantification

Lithium heparin microtainer tubes containing 0.2–0.3 ml blood were taken into a biosafety cabinet and $200\,\mu L$ of heparinized whole blood from each NHP was added to an individually labeled round-bottomed polystyrene tube (Corning/Falcon, Corning, NY, USA). Nonspecific antigen binding to Fc receptors was inhibited via 10 minutes of

blocking with Human TruStain FcX (BioLegend, San Diego, CA, USA). Cells were subsequently stained with mouse anti-human CD20 (clone 2H7, Alexa-700, catalog number 560631, lot number 0209371, 1:16 dilution), mouse anti-human CD3 (clone SP34-2, Alexa-700, catalog number catalog number 557917, lot number 9185577, 1:16 dilution), mouse anti-human CD14 (clone M5E2, Alexa-700, catalog number 557923, lot number 0023216, 1:8 dilution), mouse anti-human CD16 (clone 3G8, PE, catalog number 556619, lot number 9107538, 1:18 dilution) (all antibodies purchased from BD Biosciences and used at the manufacturer recommended dilution), and Live/Dead Fixable Blue (Invitrogen, Waltham, MA, USA) in a volume of 100 µL for 1 hour in the dark with gentle agitation every 15 minutes. Subsequently, erythrocytes were lysed with BD FACS Lysing Solution (BD Biosciences) diluted in molecular biology grade water (Corning, Corning, NY, USA) for 12 min followed by immediate centrifugation at 500 g for 5 minutes at room temperature. The supernatant was immediately aspirated, and the cellular pellet was reconstituted in FACS Buffer (0.2% bovine serum albumin, 2 mM EDTA in phosphate-buffered saline). Samples were centrifuged at 350 g for 5 minutes at room temperature. FACS buffer was removed, and pellets were fixed in freshly prepared 2% methanol-free formaldehyde (diluted from 16% Ultra Pure EM grade formaldehyde (Polysciences, Warrington, PA, USA) for a minimum of 1 h prior to analysis. Analyses were conducted using a BD LSRFortessa Cell Analyzer (BD, Franklin Lakes, NJ, USA), and $10^5$ total cells were analyzed. Natural killer cells were defined as the live cell population that were $CD16^+$ and $CD3^-CD14^-CD20^-$ (see gating strategy in Fig. S7). This population is functionally analogous to human $CD56^{dim}$ NK cells, characterized by high cytolytic activity and low cytokine production[77]. Data analysis was conducted utilizing FlowJo software (FlowJo, Ashland, OR, USA).

### Plaque reduction neutralization titers

PRNTs were performed in Vero cells in a 12-well tissue culture plate format[78]. Pre-exposure serum and 28-day post-exposure serum samples were heat inactivated in a 56 °C water bath for 60 minutes to inactivate complement. Serum samples subsequently underwent 2-fold serial dilution series in DMEM with 2% FBS and 1% P/S (1:10 to 1:320). An equal volume (75 µl) of the designated virus at a titer of 800 focus forming units (FFU) per milliliter was added to each serum dilution (dilution series range 1:20-1:640). Serum/virus mixtures were incubated for one hour at 37 °C, at which point 100 µL of mixture was added to a nearly confluent (85-95%) monolayer of Vero cells and allowed to adsorb for one hour at 37 °C. Each well was then overlayed DMEM containing 3% FBS, 1% Pen–Strep, 1.25 µL/mL amphotericin B, and 0.8% weight/vol methylcellulose and incubated 37 °C for 5 days. Fixation and immunostaining procedures took place as described above. PRNT titers were represented as the reciprocal of the highest dilution of serum that inhibited a specified percentage foci when compared to a virus only control.

### Statistical analysis and model fitting

To test a priori predictions within and between each species-virus combination, we used standard two-tailed parametric tests, non-parametric tests or generalized linear models, as appropriate. A priori power analyses were conducted to assess our ability to detect changes in peak titer and duration of titer between low and high-dose infections in macaques, resulting in a sample size of 10 in the infected group. As one macaque was transferred to the control group, and as analyses were primarily focused on NK cells and neutralizing antibody titers due to the paucity of viremia, we conducted a *post hoc* test of power, and found that, with the effect sizes and standard deviations reported, we had approximately 75% power to detect a significant difference in comparisons among 2 or 3 groups and lower power of approximately 20% for linear regressions. Sample sizes were smaller for infected squirrel monkeys (N = 8 per virus) and power was concomitantly lower.

For models testing interactions, if the interactions term was non-significant with an effect size $\eta^2 < 0.01$, we relied on a type II ANOVA for the significance of main effects, following recommendations by Smith and Cribbie[79].

The number of infectious bites initially received per NHP, as well as the viral titers contained in biting mosquitoes' saliva, are the components driving the initial dose delivered to NHPs, which can then drive their viral dynamics and immune response. Differences between experiments regarding those factors were assessed using generalized linear models (Supplementary Text S1).

To quantify a possible relationship between host infectious titer ($log_{10}$ pfu/ml) and probability to infect mosquitoes, we first used a generalized additive model. The probability of mosquito infection was broadly defined, measured by a positive mosquito body or leg. We used a binomial error distribution and constrained the number of knots to 6 to avoid overfitting. Separate relationships were fitted for DENV-2 and ZIKV, and transmission from both NHP species was considered at once.

We then refined the fitting of dose-response relationships between ZIKV infectious titers ($log_{10}$ pfu/ml) of squirrel monkeys and the probability of infecting mosquitoes. Separate relationships were fitted depending on whether mosquito infection was measured by the presence of virus in their abdomens or their legs (Supplementary Text S4). In order to compare these ZIKV dose-response relationships to existing results on dengue virus transmission from humans to *Aedes aegypti*, we retrieved data from Nguyen et al.[53] (for abdomen infections) and Duong et al.[51] (for leg infections), through the supplementary material of ten Bosch et al.[80]. As those data were in units RNA copies/ml, we applied a conversion factor, specific to each dengue serotype from Blaney et al.[81] (Supplementary Text S4), to transform it into pfu/ml, and applied the same fitting procedure. We also retrieved data from Long et al.[52] in which DENV-2 infection of mosquito bodies and heads was measured, and human viral load was in FFU/ml. We considered infection of both mosquito heads and legs as a marker of dissemination.

### Reporting summary

Further information on research design is available in the Nature Portfolio Reporting Summary linked to this article.

## Data availability

All raw data from macaques infected with DENV-2 or ZIKV and squirrel monkeys infected with DENV-2 or ZIKV, save for temperature and PRNT values against non-infecting viruses, are provided in Supplementary Datasets 1, 2, 3, and 4. For additional data, see README and data subfolder in GitHub repository https://github.com/helenececilia/hanley_2024_sylvatic_DENV_ZIKV_trade_offs. Source data are provided with this paper.

## Code availability

All analyses were performed in R and all scripts, data, and outputs are available in a citable GitHub repository[82], available at https://github.com/helenececilia/hanley_2024_sylvatic_DENV_ZIKV_trade_offs/. R version 4.3.1 was used, and analyses relied on packages glmmTMB 1.1.7, DHARMa 0.4.6, multcomp 1.4-25, car 3.1-2, bbmle 1.0-25, mgcv 1.9-0, emdbook 1.3.13, tmvtnorm 1.5.

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

## Acknowledgements

The authors thank the members of the support staff at UTMB and NMSU, who worked diligently in the face of COVID-19 restrictions during the pandemic. We thank Andrew Montoya at NMSU for his assistance in titering mosquitoes. This study was funded by NIH grant R01AI145918 to K.A.H., N.V., B.M.A., and S.L.R. Squirrel monkeys acquired from the UT MD Anderson Cancer Center, Michael E. Keeling Center for Comparative Medicine and Research were supported by NIH grant 5P40OD010938-36 prior to sale to UTMB.

## Author contributions

Conceptualization: K.A.H., N.V., B.M.A., S.L.R.; Implementation of animal studies: S.R.A., S.L.R., N.V., N.I.O.d.S., R.Y., K.H.; Optimization and implementation of assays: S.R.A., B.A.M., J.T.G., S.L.R., N.V., N.I.O.d.S., R.Y., W.Y., K.H.; Data analysis: H.C., K.A.H., B.M.A.; Writing original draft: K.A.H., H.C., S.R.A., N.V.; Review and editing: K.A.H., H.C., B.M.A., N.V., S.R.A., S.L.R., B.A.M., J.T.G., N.I.O.d.S., R.Y., W.Y.; Supervision: K.A.H., N.V., S.L.R.; Funding acquisition: K.A.H., N.V., B.M.A., S.L.R.; All authors read and approved the final manuscript

## Competing interests

The authors declare no competing interests.
