## [Peer Review File · Nature Communications]

Trade-offs shaping transmission of sylvatic dengue and Zika viruses in monkey hostsREVIEWER COMMENTS

Reviewer #1 (Remarks to the Author):

Zika and dengue viruses are thought to have originated from sylvatic cycles involving nonhuman primates in Africa and Asia, respectively. It remains unclear why this has not happened in the Americas where similar ecological conditions exist. Here, Hanley et al., use mosquito-bite delivery of dengue and Zika viruses to explore the trade-offs that shape the within-host infection dynamics for these viruses in native and novel nonhuman primate hosts. They also assessed the potential of these viruses to be transmitted back to mosquitoes. Overall, this is a comprehensive study with some unique aspects (like using mosquito-bite delivery, squirrel monkeys, etc.), however, I believe there are a few scientific issues and several things that need to be addressed to facilitate understanding by the reader. Specific comments follow, but there is no line numbering, which makes commenting difficult.

Introduction and throughout: the authors are conflating "Old World Monkeys" in a way that may be confusing for some readers. The Old World Monkeys of Africa and Asia comprise something like ~130-160 different species that may or may not be susceptible to dengue and/or Zika viruses. While it has been shown experimentally that a number of different nonhuman primate species are susceptible to dengue and Zika virus infection, the exact sylvatic reservoir species for these viruses remain less clear. African green monkeys, patas monkeys, and Guinea baboons are important for sylvatic cycles in Africa, and there is serological evidence to support susceptibility of different macaque species (including *Macaca fascicularis*) to dengue virus in Malaysia—and virus isolation from cynomolgus macaques and dusky leaf monkeys in Malaysia—so cynomolgus macaques seem like a reasonable choice for dengue virus but this could be made more explicitly clear. In contrast, sylvatic reservoirs for Zika virus are even less clear, though there is serological evidence to suggest monkey infections in Africa and Asia. The authors cite the first isolation of Zika virus in Uganda, but leave out important context: that animal was a captive rhesus macaque (Asian-origin) and thus not a natural host for Zika virus in Africa. As a result, I strongly encourage the authors to provide additional context and define what they mean by "native versus novel host" and how this might be used to explore evolutionary trade-offs or inform assessment of risk for spillback in the Americas.

Similarly, it is clear that DENV-2 strain P8-1407 is sylvatic because it was isolated from a sentinel monkey in Malaysia and clusters phylogenetically with other sylvatic DENVs. In contrast, ZIKV DAK AR 41525 was isolated from a pool of *Aedes africanus* mosquitoes. Therefore, is the justification for classifying it as sylvatic because this mosquito species primarily feeds on nonhuman primates? If yes, please make this more clear in the main text. If there was some other justification, make that more clear.

First sentence describing detection of DENV-2 in saliva: It would be helpful to state that "DENV-2 was not detectable in the saliva [via plaque assay??]...." Also, please clarify that saliva samples were blind passaged and then used to estimate the dose of virus delivered. The statement "Thus, we used number of mosquitoes salivating detectable virus to estimate total dose of virus delivered" is confusing since the previous sentence states that no virus was detectable in the saliva.

The way seroconversion is defined is awkward, seroconversion is typically defined as greater than or equal to some PRNT50 titer, so here it would be >20 or above the limit of detection (1:20).

In the results section please provide brief additional context to orient the reader when referring to hypotheses that were introduced in the introduction.

Results: why were Zika virus infections not done in cynomolgus macaques? I acknowledge that you can't do everything but again I believe this should be more explicit. After reading the introduction and abstract, I was under the impression that these experiments were going to be included.

Results and discussion: the authors suggest that DENV and ZIKV follow different strategies in

squirrel monkeys (novel host) because viremia kinetics are different, but couldn't this simply be the result of the inherent capacity of this ZIKV strain to replicate better overall, regardless of host species? In fact, the authors note that DENV-2 infection parameters did not differ between cynos and squirrel monkeys. What is known about this ZIKV strain and others's replication kinetics in macaques (which I acknowledge one could argue are novel hosts). Similarly, how does sylvatic DENV-2 infection in cynomolgus macaque compare with non-sylvatic DENV-2 infections in macaques (again acknowledging that those experiments were in all likelihood done with needle inoculation).

Section that starts "we predicted that transmission from novel hosts would be low relative to native hosts": I think this section could be improved by a few additional sentences providing more context about genotype-by-genotype interactions for virus mosquito combinations; and while *Aedes albopictus* seems like a reasonable choice for these experiments and a likely bridge vector in nature, there may be other mosquito species that are able to become more efficiently infected at these low viremia concentrations.

Reference 30 used rhesus macaques not cynomolgus macaques for mosquito-delivery of Zika virus. However, the authors should discuss their own study in which Zika virus was delivered via mosquito-bite to cynomolgus macaques: PMID 30469417.

The reference list only goes to 49 but there are 56 studies cited in the text.

Text overall: when talking about transmission from monkeys, I believe it would be helpful to state something like "from monkeys to mosquitoes" to improve readability. For example, the statement "the rarity of transmission of sylvatic DENV-2 from squirrel monkeys suggests that the virus cannot gain a foothold in neotropical species, albeit testing more strains of mosquitoes and monkeys, as well as modeling, are still needed" could use some additional qualifiers. Does "the virus cannot gain a foothold in neotropical species" refer to nonhuman primates, mosquitoes, or both? If both, this is not entirely accurate because we know DENV-2 is efficiently transmitted by neotropical *Aedes aegypti*.

Respectfully, I am not sure I agree with this statement: "empirical data suggest that establishment of a neotropical, enzootic cycle is substantially more likely for ZIKV than DENV." This study was done with a single DENV-2 and single ZIKV strain both of which may or may not be generalizable. In addition, regarding spillback into sylvatic cycles in the Americas, wouldn't the more relevant experiment be to assess the potential for human-endemic Zika and dengue viruses from the Americas to replicate in nonhuman primates or other putative reservoir species to assess the relative likelihood for each one to establish sylvatic cycles?

Methods: were challenge stocks sequence authenticated before being used in nonhuman primate experiments?

Figure 2: blue shade bars indicate the mean number of mosquitoes infected after feeding on virus-infected monkeys BUT it states that the % of infected is derived from virus positive, legs, or saliva. I am confused about what this actually represents. Were there saliva or leg samples that were virus positive but did not have a corresponding body sample that was virus positive? Please clarify.

Reviewer #2 (Remarks to the Author):

General:

Interesting experiments addressing an important question as it relates to emergence and re-emergence of mosquito borne diseases. There are a number of factors which could impact the cycle being proposed (mosquito - NHP - mosquito) which could have (not conclusive) impacted the outcome of the study. A few of these issues include:

- mosquito and virus mismatch from an origin perspective (infectivity of mosquitoes collected from one area with viruses from another, does this impact infection and transmissibility dynamics [co-

evolution advantages])

- imbalance in male and female NHPs (does this make a difference despite similar weight)
- use of anesthesia (can this itself being immunomodulating, were controls under same anesthesia duration and conditions which would have limited this potential bias)
- importance of mosquito salivary proteins in the infection process and can this vary by mosquito making the use of small mosquito numbers important
- use of 1 versus 10 mosquitoes as a reflection of dose delivered, is this an appropriate simulation, likely not in the field
- passage of viruses and potential for mutation and its impact on infectivity
- standardization of saliva collection methods and use of it to make claims about dose delivered

Abstract:

Some of the language being used I envision will be less accessible to the general reader. Consider using more descriptive language and less terminology.

Introduction:

"but DENV has never spilled back into a sylvatic reservoir"

- Is this confirmed or has not been identified.

Results:

"delivered by infected *Ae. albopictus* mosquitoes"

- Do the authors have thoughts on how using *Ae. aegypti* would have affected the experiments. "Malaysian strain of DENV-2"
- Data exists supporting differences in infectivity depending on origin of mosquito and virus species/strains (co-evolution). How does the mosquito strain used align with the virus strain used (geographic source)?

Discussion:

Methods:

"viruses with the following passage histories"

- Is the passage impact on genetic mutation understood and how this may impact infectivity and replication potential?

"was placed on the anesthetized animal's ear"

- The process of anesthesia (stress) may be immunomodulating prompting some groups to use behavioral training to complete NHP studies. What are the authors' opinions on this matter?

"virus only control"

- What were the virus controls (same as screening controls)? If different than viruses used to infect animals how could this impact data and conclusions?

Table 1:

- There does not appear to be balance between use of Males and Females. Could this make a difference even if weights are similar?

Reviewer #3 (Remarks to the Author):

In this report, Hanley et al, examined DENV and ZIKV infection of old and new world NHP to understand viral replication and transmission in the context of mosquito transmission. They used a mixture of experimental and in silico approaches to test 3 hypotheses. In the study, the investigators made a few important observations, most notably that NHP to mosquito transmission of DENV occurred exclusively when serum viral titer was undetectable or nearly so. Otherwise, the study is a series of observations that hint at interesting findings but are largely underpowered, contradictory with other data, or are based on simplistic assumptions. The authors sought to test three distinct and potentially important hypotheses, but it's not clear which, if any, of the hypotheses are best supported by the data. Some examples of issues with the manuscript are as follows.

Cynos were more likely to become infected (only DENV tested) as dose increased while squirrel monkeys were not (with either virus) (hyp 1), which may be informative, but cynos were not infected with Zika preventing a complete comparison, and there are several important caveats, primarily the very small number of observed transmission events, which is addressed. Thus, it's not clear how meaningful these data really are.

Regarding hypothesis 2; the authors show that there is an inverse relationship between magnitude of NK cell influx and nAb titer (though this relationship is not significant in most comparisons) and propose NK cells might even mediate viral control, leading to lower nAb titers. Though this may be true, the data presented are insufficient to make this claim. NK cell frequency may simply track another antiviral or inflammatory response and have no causative role in viral control (and the authors even recognize this caveat in the discussion). Further, NK cells are a highly heterogenous cell type and no possible mechanisms are proposed or demonstrated. It's not clear if these data have any relevance to viral control.

Regarding hypothesis 3, the authors acknowledge that the very low number of transmission events from either monkey species to mosquitos renders it unclear if indeed transmission from novel hosts is less efficient than from native hosts. So this hypothesis isn't ever actually tested.

Notably, data from cytokines was inconclusive and not informative for hypothesis testing.

All in all, there just doesn't appear to be a singular message that can be gleaned from the data collected.

Reviewer #4 (Remarks to the Author):

Review of "Immunologically mediated trade-offs shaping transmission of sylvatic dengue and Zika viruses in native and novel non-human primate hosts" by Hanley et al. submitted to Nature Communications (NCOMMS-23-29346).

This MS reports on a study whether insights in the factors that govern host exploitation trade-offs can help to predict how arthropod-borne viruses fare in novel hosts. This is mainly an experimental study, involving two arbo viruses (a Dengue strain and Zika) and two primate host species, one of which is an original host and the other a novel host. The study is set up so as to assess infection success and subsequent transmission parameters (viremia, duration of infectivity, mosquito infection rate).

Let me begin by stating that I am not qualified to assess the experimental protocol and also that statistics is not my strongest point either. But I found the study very interesting and the text well-structured. Among the results that struck me most was the finding that there doesn't seem to be a clear relationship between virus load and transmission to mosquitos, something I tend to take for granted. This doesn't have to do much with the original question the authors pose but it certainly stimulates thinking of what might underlie this.

One of the problems with this kind of study is the low numbers of experimental animals involved. Due to the inherent limitations such experimental studies impose it is very important that the hypotheses to be tested be carefully formulated to maximise the chances of obtaining significant results. It is here that I found the MS the most wanting.

The MS begins by pointing out that human migration can introduce pathogens in novel ecosystems, and that this has actually happened in the past. Given the current massive scale of global travel it is likely to happen more often nowadays, so we have to know what determines what governs the chances of establishment of a pathogen in a new environment. The introduction then argues that this depends in part on whether the trade-offs that the pathogen is subject to in

its novel host are similar to the ones in its original host. I quite agree with this, and indeed any insight into what governs ANY such tradeoff is most welcome (in particular the transmission-clearance trade-off, as is the case here).

The introduction then goes on to formulate three different hypotheses that the study will address. In my view, however, these `hypotheses' are not hypotheses, but rather predictions resulting from a common hypothesised process, namely that the transmission-clearance trade-off results from the interaction of the virus with the host's immune system. (Even if they were true hypotheses, I don't think the low sample sizes involved in the study would allow clear rejection of any of these anyway).

Again, I'm not qualified to assess the experimental setup and protocols but it seems well designed. And again, the observation that viremia doesn't seem to be linked to transmission success is very puzzling and certainly worthy of publication. Also the fact that there seem to be differences in the effects of the same pathogen on different hosts is interesting, but this could be due to so many other factors than that one is a novel host and the other an original host that this does not corroborate or refute (what I see as) the original hypothesis.

Coming back to this original hypothesis, I want to make a number of remarks, and argue that these necessitate at least a substantial rewrite.

- First and foremost, the notion of a transmission-duration trade-off does not, in itself, allow one to distinguish among original and novel hosts. The idea behind such a trade-off is that a well-adapted strain cannot increase transmission without shortening duration or vice versa. A virus that finds itself in a host to which it is not fully adapted can conceivably increase BOTH traits at the same time until it hits the boundary of what is possible, after which it is subject to a similar trade-off (which may be host-specific). The only way I see to study this experimentally is by monitoring the process of adaptation, which is hard to do and what this study definitely did not do.

- Second, while it is conceivable that increasing infective dose may shorten the time to peak viremia, I wouldn't bet a dollar on it that this effect would be detected in the current setup (unless the difference was in a multitude of orders of magnitude). Where infective dose MAY be important is in the probability of establishment of the infection but that doesn't seem to be an issue here, neither in the original nor in the novel host (I would not have been surprised at all if the viruses would have difficulty infecting the novel hosts).

- I'm not an immunologist, so I had considerable difficulty in interpreting the various immunological parameters that are measured. In the very least indicate which of these are linked to the innate immune system and which to the adaptive immune system as both affect clearance differently.

To sum it up, it was not really clear to me how the study could assess the difference of virus performance between original and novel hosts nor what differences one would expect in immunological responses. And apparently mosquitoes are way more sensitive to the viruses than the authors' ability to detect them...

Minor things

- p.6 "Sylvatic ZIKV replicated to high levels in squirrel monkeys, stimulating low neutralizing antibody titers" Stimulating? I don't get this. Isn't this more indicative of low antibody titers?

- p.7 Two squirrel monkeys had to be euthanised? I agree that one cannot attribute the reasons for this to them being infected with Zika with certainty, but one cannot exclude it either! Zika is known to cause neurological symptoms and it may cause arthritis. Whatever the cause, a mortality rate of 2/30 in infected animals (if I'm correct) is significant.

On behalf of my coauthors, I thank our four reviewers for their careful consideration and thoughtful critiques of our manuscript (NCOMMS-23-29346). We have now added new data to the manuscript (Zika virus infection of cynomolgus macaques) and substantially revised the organization of the results and discussion.

Please see our point-by-point responses to review below. Please note that we have included two additional authors who led the generation of the new data.

Reviewer 1

C1.1: Zika and dengue viruses are thought to have originated from sylvatic cycles involving nonhuman primates in Africa and Asia, respectively. It remains unclear why this has not happened in the Americas where similar ecological conditions exist. Here, Hanley et al., use mosquito-bite delivery of dengue and Zika viruses to explore the trade-offs that shape the within-host infection dynamics for these viruses in native and novel nonhuman primate hosts. They also assessed the potential of these viruses to be transmitted back to mosquitoes. Overall, this is a comprehensive study with some unique aspects (like using mosquito-bite delivery, squirrel monkeys, etc.), however, I believe there are a few scientific issues and several things that need to be addressed to facilitate understanding by the reader. Specific comments follow, but there is no line numbering, which makes commenting difficult.

R1.1. We apologize for the oversight and have added line numbering.

*C1.2: Introduction and throughout: the authors are conflating “Old World Monkeys” in a way that may be confusing for some readers. The Old World Monkeys of Africa and Asia comprise something like ~130-160 different species that may or may not be susceptible to dengue and/or Zika viruses. While it has been shown experimentally that a number of different nonhuman primate species are susceptible to dengue and Zika virus infection, the exact sylvatic reservoir species for these viruses remain less clear. African green monkeys, patas monkeys, and Guinea baboons are important for sylvatic cycles in Africa, and there is serological evidence to support susceptibility of different macaque species (including *Macaca fascicularis*) to dengue virus in Malaysia—and virus isolation from cynomolgus macaques and dusky leaf monkeys in Malaysia—so cynomolgus macaques seem like a reasonable choice for dengue virus but this could be made more explicitly clear. In contrast, sylvatic reservoirs for Zika virus are even less clear, though there is serological evidence to suggest monkey infections in Africa and Asia. The authors cite the first isolation of Zika virus in Uganda, but leave out important context: that animal was a captive rhesus macaque (Asian-origin) and thus not a natural host for Zika virus in Africa. As a result, I strongly encourage the authors to provide additional context and define what they mean by “native versus novel host” and how this might be used to explore evolutionary trade-offs or inform assessment of risk for spillback in the Americas.*

R1.2: We appreciate the reviewer’s call for more detail; we have now added the additional context in line 59 and lines 92-102.

C1.3. Similarly, it is clear that DENV-2 strain P8-1407 is sylvatic because it was isolated from a sentinel monkey in Malaysia and clusters phylogenetically with other sylvatic

DENVs. In contrast, ZIKV DAK AR 41525 was isolated from a pool of Aedes africanus mosquitoes. Therefore, is the justification for classifying it as sylvatic because this mosquito species primarily feeds on nonhuman primates? If yes, please make this more clear in the main text. If there was some other justification, make that more clear.

R1.3. We have now added some additional information about the natural history of these strains to justify their designation as sylvatic in lines 105-108.

C1.4. First sentence describing detection of DENV-2 in saliva: It would be helpful to state that "DENV-2 was not detectable in the saliva [via plaque assay??]...." Also, please clarify that saliva samples were blind passaged and then used to estimate the dose of virus delivered. The statement "Thus, we used number of mosquitoes salivating detectable virus to estimate total dose of virus delivered" is confusing since the previous sentence states that no virus was detectable in the saliva.

R1.4. We have made the requested clarifications in lines 137-139.

C1.5: The way seroconversion is defined is awkward, seroconversion is typically defined as greater than or equal to some PRNT50 titer, so here it would be >20 or above the limit of detection (1:20).

R1.5. We have made the requested clarifications in lines 145 and 150.

C1.6: In the results section please provide brief additional context to orient the reader when referring to hypotheses that were introduced in the introduction.

R1.6. We have reorganized the results section quite extensively to more clearly link hypotheses and results.

C1.7: Results: why were Zika virus infections not done in cynomolgus macaques? I acknowledge that you can't do everything but again I believe this should be more explicit. After reading the introduction and abstract, I was under the impression that these experiments were going to be included.

R1.7: We had difficulty obtaining enough cynomolgus macaques for this study due to the pandemic (availability went down and price went up dramatically). However, after the completion of the experiments originally described we were able to obtain three additional cynomolgus macaques that we infected with sylvatic Zika virus as in the rest of the study. This reviewer's comments spurred us to accelerate analysis of these samples, and this data is now included in the study. Essentially, the dynamics of ZIKV in macaques were similar to dynamics in squirrel monkeys, save that ZIKV ID50 was lower in squirrel monkeys than macaques, and macaques produced robust neutralizing antibody titers while squirrel monkeys did not.

C1.8. Results and discussion: the authors suggest that DENV and ZIKV follow different strategies in squirrel monkeys (novel host) because viremia kinetics are different, but couldn't this simply be the result of the inherent capacity of this ZIKV strain to replicate better overall, regardless of host species? In fact, the authors note that DENV-2 infection parameters did not differ between cynos and squirrel monkeys. What is known about this ZIKV strain and others's replication kinetics in macaques (which I acknowledge one could argue are novel hosts). Similarly, how does sylvatic DENV-2 infection in

cynomolgus macaque compare with non-sylvatic DENV-2 infections in macaques (again acknowledging that those experiments were in all likelihood done with needle inoculation).

R1.8. We have included comparison of replication of ZIKV in our study with replication of other ZIKV strains delivered by needle and mosquito bite to macaques and neotropical primates in the discussion (lines 461-468) and to replication of the same strain of ZIKV in *Ae. aegypti* and *Ae. albopictus* (lines 395-398). Moreover, we have added an explicit comparison of the current data with our study of African sylvatic DENV-2 (the only other study of sylvatic DENV in primates of which we are aware, lines 418-421). Generally, we find consistent patterns of replication among these studies, suggesting that the dynamics that we detected are not strain specific and indeed that mosquito delivery does not greatly alter dynamics. That said, studies like ours using other ZIKV and DENV strains will be welcome, and indeed are underway in our labs.

C1.9. Section that starts “we predicted that transmission from novel hosts would be low relative to native hosts”: I think this section could be improved by a few additional sentences providing more context about genotype-by-genotype interactions for virus mosquito combinations; and while Aedes albopictus seems like a reasonable choice for these experiments and a likely bridge vector in nature, there may be other mosquito species that are able to become more efficiently infected at these low viremia concentrations.

R1.9. We have added a caveat to this effect (lines 449 -452) and also please see R1.8 for evidence that the infectivity of our ZIKV strain seems to be quite generalizable across strains of *Ae. aegypti* and *Ae. albopictus*. We note, just for the reviewer’s information, that we did attempt to feed *Sabethes cyaneus*, the only sylvatic, neotropical yellow fever virus vector held in colony, on our monkeys. Unfortunately, this species initiates feeding so slowly that it was not possible to hold the monkeys under anesthesia long enough to enable feeding.

C1.10. Reference 30 used rhesus macaques not cynomolgus macaques for mosquito-delivery of Zika virus. However, the authors should discuss their own study in which Zika virus was delivered via mosquito-bite to cynomolgus macaques: PMID 30469417.

R1.10. In the study referenced above, cynomolgus macaques were infected via subcutaneous inoculation and then uninfected mosquitoes were fed upon them to assess transmission. Thus, we did not cite it in the sentence referring to the effects of mosquito delivery of virus, although it is included in the Discussion.

C1.11. The reference list only goes to 49 but there are 56 studies cited in the text.

R1.11. Apologies for this error; it is now rectified.

C1.12. Text overall: when talking about transmission from monkeys, I believe it would be helpful to state something like “from monkeys to mosquitoes” to improve readability. For example, the statement “the rarity of transmission of sylvatic DENV-2 from squirrel monkeys suggests that the virus cannot gain a foothold in neotropical species, albeit testing more strains of mosquitoes and monkeys, as well as modeling, are still needed” could use some additional qualifiers. Does “the virus cannot gain a foothold in neotropical species” refer to nonhuman primates, mosquitoes, or both? If both, this is not

entirely accurate because we know DENV-2 is efficiently transmitted by neotropical Aedes aegypti.

R1.12. We have clarified throughout.

C1.13. Respectfully, I am not sure I agree with this statement: “empirical data suggest that establishment of a neotropical, enzootic cycle is substantially more likely for ZIKV than DENV.” This study was done with a single DENV-2 and single ZIKV strain both of which may or may not be generalizable. In addition, regarding spillback into sylvatic cycles in the Americas, wouldn’t the more relevant experiment be to assess the potential for human-endemic Zika and dengue viruses from the Americas to replicate in nonhuman primates or other putative reservoir species to assess the relative likelihood for each one to establish sylvatic cycles?

R1.13. We agree with the reviewer that similar studies with human-endemic strains of ZIKV and DENV are needed, and indeed these are ongoing in our laboratories, but they are outside the scope of the current manuscript. We disagree that sylvatic DENV or ZIKV are unlikely to be translocated to the Americas; sylvatic DENV has been detected in humans that had been infected in Africa and Asia and then traveled to Europe and Australia, respectively. We are not aware of similar detections of sylvatic ZIKV, but we suspect this has more to do with lack of surveillance than lack of translocation. We have added a few words to this effect in the Introduction, lines 49-50.

C1.14. Methods: were challenge stocks sequence authenticated before being used in nonhuman primate experiments?

R1.14. Yes, this is standard practice in our laboratories.

C1.15. Figure 2: blue shade bars indicate the mean number of mosquitoes infected after feeding on virus-infected monkeys BUT it states that the % of infected is derived from virus positive, legs, or saliva. I am confused about what this actually represents. Were there saliva or leg samples that were virus positive but did not have a corresponding body sample that was virus positive? Please clarify.

C1.16. As now noted in the results, for both DENV and ZIKV, virus was often detected in the legs more often than the body (lines 177-179). We have looked in the literature for other studies that may have found a similar pattern, but we find that, very often, only legs from infected bodies are tested, making it impossible to detect infected legs in the absence of an infected body. Older studies of DENV replication in *Aedes aegypti*¹ have shown that DENV titer decreases in the midgut rather rapidly post-infection but stays high in the haemolymph, and we believe these dynamics likely explain the difference in detection in our studies. We have removed “saliva” from the legend since no mosquito was saliva positive if it was not also leg positive.

Reviewer 2

C2.1: General:

Interesting experiments addressing an important question as it relates to emergence and re-emergence of mosquito borne diseases. There are a number of factors which could impact the cycle being proposed (mosquito - NHP - mosquito) which could have (not conclusive) impacted the outcome of the study. A few of these issues include:

- mosquito and virus mismatch from an origin perspective (infectivity of mosquitoes collected from one area with viruses from another, does this impact infection and transmissibility dynamics [co-evolution advantages])

R2.1: We agree, please see R1.9.

C2.2 - imbalance in male and female NHPs (does this make a difference despite similar weight)

R2.2. Actually, we used a stratified randomization scheme to ensure that male and female monkeys were balanced across treatments, although of course when there is an odd number of monkeys there will be more of one or another sex. The balance is the following:

Cynomogus macaques

Control: 2M, 2F

DENV low: 2M, 2F

DENV high: 2M, 3F

ZIKV: 1M, 2F

Squirrel monkeys

Control: 2M, 2F

DENV: 5M, 5F

ZIKV: 5M, 5F

C2.3 use of anesthesia (can this itself being immunomodulating, were controls under same anesthesia duration and conditions which would have limited this potential bias)

R2.3. Controls were treated exactly the same as infected animals save that the mosquitoes that fed upon controls on D0 were not infected.

C2.4: importance of mosquito salivary proteins in the infection process and can this vary by mosquito making the use of small mosquito numbers important.

R2.4: We agree with the reviewer that salivary gland proteins have the potential to impact virus dynamics. However, for the portion of the experiment where we deliberately used a low or high dose delivery, we saw scanty viremia but no pattern with regard to dose (i.e. we detected viremia following both low and high dose infections).

C2.5: use of 1 versus 10 mosquitoes as a reflection of dose delivered, is this an appropriate simulation, likely not in the field.

R2.5: As we do not know how many infectious bites a monkey may receive in a field situation, we cannot answer this comment with data, however we do note that dose had relatively little impact on virus dynamics. We certainly think that delivery of virus by mosquito is a closer approximation to natural conditions than needle delivery.

C2.6: passage of viruses and potential for mutation and its impact on infectivity

R2.6: We have added this caveat to the discussion in line 449.

C2.7: standardization of saliva collection methods and use of it to make claims about dose delivered.

R2.7: We are not sure that we understand this comment; our collection of saliva is completely standardized, and all mosquitoes salivate into a collection tube for exactly the same amount of time. We included as a caveat that forced salivation likely represents relative dose rather than actual dose delivered during authentic feeding, lines 369-371.

C2.8: Abstract: Some of the language being used I envision will be less accessible to the general reader. Consider using more descriptive language and less terminology.

R2.8: While the tight word limit on the abstract leaves little space for explanation of terms, we have now noted that sylvatic cycles occur in the forest in lines 22-23. hopefully that clarifies what may be the least accessible term in the abstract.

*C2.9: Introduction: "but DENV has never spilled back into a sylvatic reservoir"
- Is this confirmed or has not been identified.*

R 2.9: It is impossible to confirm a negative, but studies have sought evidence of a sylvatic cycle in the Americas and failed to find one (Turell et al. 2019 and references therein), and we have now clarified this sentence and added this reference in line 44.

*C2.10. Results: "delivered by infected Ae. albopictus mosquitoes"
- Do the authors have thoughts on how using Ae. aegypti would have affected the experiments.*

R2.10. We chose to use *Ae. albopictus* because this species, compared to *Ae. aegypti*, is much more likely to be involved in sylvatic transmission and spillover. However, we suspect that results using *Ae. aegypti* would have been similar, particularly for transmission. Multiple studies have previously shown similar infection, dissemination and salivation of ZIKV 41525 in both *Ae. albopictus* and *Ae. aegypti* using multiple infection paradigms and multiple strains of each species²⁻⁴, and we have added this information to our discussion in lines 395-397. Similar experiments have not been conducted yet for sylvatic DENV.

*C2.11, "Malaysian strain of DENV-2"
- Data exists supporting differences in infectivity depending on origin of mosquito and virus species/strains (co-evolution). How does the mosquito strain used align with the virus strain used (geographic source)?*

R2.11. As discussed above and in our original manuscript, the potential effects of a GxG mismatch in our study is an important caveat to our study. However, one of our concerns is spillback, in which the virus and mosquito species will have no co-evolutionary history.

C2.12: Discussion:

R1.12: We note that the reviewer had no comments on our discussion.

*C2.13. Methods: "viruses with the following passage histories"
- Is the passage impact on genetic mutation understood and how this may impact infectivity and replication potential?*

R2.12. See R2.6

C2.14. "was placed on the anesthetized animal's ear"

- The process of anesthesia (stress) may be immunomodulating prompting some groups to use behavioral training to complete NHP studies. What are the authors opinions on this matter?

R2.14. Since the container had to be held by an investigator on a monkey's ear, and therefore in close proximity to its mouth, for a prolonged period of time, our opinion (and that of our IACUC) was that anesthesia was necessary to ensure investigator safety.

C2.15. "virus only control"

- What were the virus controls (same as screening controls)? If different than viruses used to infect animals how could this impact data and conclusions?

R2.15. The reviewer is referring to our description of the implementation of PRNT assays; the point here is that we calculate the diminution in number of plaques of a given virus in wells that contain a certain dilution of serum relative to wells that contain only that same virus and no serum. We use the panel of viruses listed in the methods for our PRNT assays as they have been well characterized for this assay; since ZIKV comprises a single serotype and our previous research has shown broad neutralization of DENV across serotypes⁵, including sylvatic strains, this should not affect the results.

C2.16. Table 1:

- There does not appear to be balance between use of Males and Females. Could this make a difference even if weights are similar?

R2.16. See R2.2

Reviewer 3

C3.1. In this report, Hanley et al, examined DENV and ZIKV infection of old and new world NHP to understand viral replication and transmission in the context of mosquito transmission. They used a mixture of experimental and in silico approaches to test 3 hypotheses. In the study, the investigators made a few important observations, most notably that NHP to mosquito transmission of DENV occurred exclusively when serum viral titer was undetectable or nearly so. Otherwise, the study is a series of observations that hint at interesting findings but are largely underpowered, contradictory with other data, or are based on simplistic assumptions.

R3.1. We have added substantial new data to the paper and reorganized the results and discussion for additional clarity; we hope that this will convince the reviewer of the study's merit.

C3.2. The authors sought to test three distinct and potentially important hypotheses, but it's not clear which, if any, of the hypotheses are best supported by the data.

R3.2. We have restructured the results and discussion to more clearly demonstrate the relevance of the data to the initial hypotheses.

C3.3. Some examples of issues with the manuscript are as follows.

Cynos were more likely to become infected (only DENV tested) as dose increased while squirrel monkeys were not (with either virus) (hyp 1), which may be informative, but cynos were not infected with Zika preventing a complete comparison,

R3.3. We have now added data for cynos infected with Zika virus, enabling a complete comparison; please see R1.7 for a more complete explanation

C3.4. and there are several important caveats, primarily the very small number of observed transmission events, which is addressed. Thus, it's not clear how meaningful these data really are.

R3.4. We disagree with the reviewer that these data are not meaningful; please see our newly discussed comparison to several key studies in humans (one of which was published while this manuscript was in review) in lines 379-388.

C3.5. Regarding hypothesis 2; the authors show that there is an inverse relationship between magnitude of NK cell influx and nAb titer (though this relationship is not significant in most comparisons) and propose NK cells might even mediate viral control, leading to lower nAb titers. Though this may be true, the data presented are insufficient to make this claim. NK cell frequency may simply track another antiviral or inflammatory response and have no causative role in viral control (and the authors even recognize this caveat in the discussion). Further, NK cells are a highly heterogenous cell type and no possible mechanisms are proposed or demonstrated. It's not clear if these data have any relevance to viral control.

R3.5. We have been careful to avoid claiming that NK cells themselves are causative in viral control, and we have added language throughout the manuscript to make this more clear (e.g. lines 82-83). We agree that NK cells are highly heterogenous, but unfortunately the tools for teasing out such heterogeneity in macaques are not fully mature yet. However, we have added text to the methods to indicate that the population of NK cells that we identify here are equivalent to CD56 dim cells in humans, which are known to be involved in antiviral response (lines 639-640). Furthermore, we focused on NK cells because this is the cell type used in at least one key model of within-host replication of arboviruses, and thus empirical data on this cell type will be of great interest.

C3.6. Regarding hypothesis 3, the authors acknowledge that the very low number of transmission events from either monkey species to mosquitos renders it unclear if indeed transmission from novel hosts is less efficient than from native hosts. So this hypothesis isn't ever actually tested.

R3.6. We have now added data for Zika virus infection of cynomolgus macaques that reveals that transmission of this virus from squirrel monkeys is actually more efficient (lower ID50) rather than less efficient than transmission from cynos. We believe this offers a strong contradiction of hypothesis 3.

C3.7. Notably, data from cytokines was inconclusive and not informative for hypothesis testing.

R3.7. We agree and have removed the cytokine data from the paper. We used highlighting rather than “track changes” to show changes to the paper since we find this easier to see, so the removal of the cytokine data is not tracked in the manuscript.

C3.8. All in all, there just doesn't appear to be a singular message that can be gleaned from the data collected.

R3.8. We hope that, in light of the additional data and additional clarifications that we have added, the reviewer will be more convinced of the value of our study.

Reviewer 4

C4.1. Review of "Immunologically mediated trade-offs shaping transmission of sylvatic dengue and Zika viruses in native and novel non-human primate hosts" by Hanley et al. submitted to Nature Communications (NCOMMS-23-29346).

This MS reports on a study whether insights in the factors that govern host exploitation trade-offs can help to predict how arthropod-borne viruses fare in novel hosts. This is mainly an experimental study, involving two arbo viruses (a Dengue strain and Zika) and two primate host species, one of which is an original host and the other a novel host. The study is set up so as to assess infection success and subsequent transmission parameters (viremia, duration of infectivity, mosquito infection rate).

Let me begin by stating that I am not qualified to assess the experimental protocol and also that statistics is not my strongest point either. But I found the study very interesting and the text well-structured. Among the results that struck me most was the finding that there doesn't seem to be a clear relationship between virus load and transmission to mosquitos, something I tend to take for granted. This doesn't have to do much with the original question the authors pose but it certainly stimulates thinking of what might underlie this.

One of the problems with this kind of study is the low numbers of experimental animals involved. Due to the inherent limitations such experimental studies impose it is very important that the hypotheses to be tested be carefully formulated to maximise the chances of obtaining significant results. It is here that I found the MS the most wanting.

R4.1. We note that we conducted careful analyses of power prior to conducting this study, and that we did in fact detect numerous statistically significant associations as predicted by our hypothesis. Moreover, we emphasize that our analysis of transmission is highly powered due to repeat sampling across days.

C4.2 The MS begins by pointing out that human migration can introduce pathogens in novel ecosystems, and that this has actually happened in the past. Given the current massive scale of global travel it is likely to happen more often nowadays, so we have to know what determines what governs the chances of establishment of a pathogen in a new environment. The introduction then argues that this depends in part on whether the trade-offs that the pathogen is subject to in its novel host are similar to the ones in its original host. I quite agree with this, and indeed any insight into what governs ANY such tradeoff is most welcome (in particular the transmission-clearance trade-off, as is the case here).

R4.2. We thank the reviewer for their kind words.

C4.3. The introduction then goes on to formulate three different hypotheses that the study will address. In my view, however, these 'hypotheses' are not hypotheses, but rather predictions resulting from a common hypothesised process, namely that the transmission-clearance trade-off results from the interaction of the virus with the host's immune system. Even if they were true hypotheses, I don't think the low sample sizes involved in the study would allow clear rejection of any of these anyway).

Again, I'm not qualified to assess the experimental setup and protocols but it seems well designed. And again, the observation that viremia doesn't seem to be linked to transmission success is very puzzling and certainly worthy of publication.

R4.3. Respectfully we disagree that the study addresses a single hypothesis. We would argue that our first hypothesis is that a transmission-clearance trade-off occurs (and implicit in the name is that this is mediated by some aspect of immunity), our second hypothesis hones in on a specific mechanism mediating this trade-off, namely natural killer cells, and our third hypothesis invokes differences among host species based on their co-evolutionary history with these two viruses. We have made these distinctions clearer in lines 76-78 and we have retained the initial structure of three separate hypotheses in the introduction and throughout the manuscript.

C4.4. Also the fact that there seem to be differences in the effects of the same pathogen on different hosts is interesting, but this could be due to so many other factors than that one is a novel host and the other an original host that this does not corroborate or refute (what I see as) the original hypothesis.

R4.4.. We believe that the addition of data from ZIKV-infected cynomolgus macaques, and the similarity in infection and dynamics between the two, strengthens the native-novel host comparison.

C4.5. Coming back to this original hypothesis, I want to make a number of remarks, and argue that these necessitate at least a substantial rewrite. First and foremost, the notion of a transmission-duration trade-off does not, in itself, allow one to distinguish among original and novel hosts. The idea behind such a trade-off is that a well-adapted strain cannot increase transmission without shortening duration or vice versa. A virus that finds itself in a host to which it is not fully adapted can conceivably increase BOTH traits at the same time until it hits the boundary of what is possible, after which it is subject to a similar trade-off (which may be host-specific). The only way I see to study this experimentally is by monitoring the process of adaptation, which is hard to do and what this study definitely did not do.

R4.5. We agree with the reviewer that a virus infecting a novel host is likely to be subject to substantial selective pressures to optimize transmission; our hypothesis (which was refuted by the data) was that it was unlikely that a virus in a new host would "hit the mark" on transmission at first contact, and that therefore we would see less efficient transmission from novel than native hosts.

C4.6. Second, while it is conceivable that increasing infective dose may shorten the time to peak viremia, I wouldn't bet a dollar on it that this effect would be detected in the

current setup (unless the difference was in a multitude of orders of magnitude). Where infective dose MAY be important is in the probability of establishment of the infection but that doesn't seem to be an issue here, neither in the original nor in the novel host (I would not have been surprised at all if the viruses would have difficulty infecting the novel hosts).

R4.6. We note that our previous experimental tests and pooled analyses have in fact detected a strong association between infective dose and time to viremia or peak viremia⁶⁻⁸; however we did not test lag to detectable viremia in this study. Rather we tested the association between dose and peak viremia (when appropriate) or simply likelihood of becoming detectably viremic, and well as the relationship between peak viremia and duration of viremia.

The association between number of infectious bits and likelihood of becoming detectably viremia was weakly positive for DENV in macaques but not significant for DENV in squirrel monkeys nor for ZIKV in squirrel monkeys and cynomolgus macaques. However, per our initial prediction, we did detect a significant, negative relationship between peak viremia and duration of viremia for ZIKV.

C4.7. I'm not an immunologist, so I had considerable difficulty in interpreting the various immunological parameters that are measured. In the very least indicate which of these are linked to the innate immune system and which to the adaptive immune system as both affect clearance differently.

R4.7. We have removed the cytokine data, which helps to streamline the considerations of immunology. NK cells have traditionally been thought to be part of the innate immune system, but there is growing evidence of NK cell memory⁹, which would move them to the adaptive arm of the immune system. We do not have adequate scope in this manuscript to discuss these nuances, unfortunately. Neutralizing antibody sits squarely in the repertoire of the adaptive immune system, but we feel that this will be known by Nature Communication readers.

C4.8. To sum it up, it was not really clear to me how the study could assess the difference of virus performance between original and novel hosts nor what differences one would expect in immunological responses. And apparently mosquitos are way more sensitive to the viruses than the authors' ability to detect them...

R4.8. We believe that the addition of ZIKV infection of cyno macaques strengthens the native-novel host comparison. We point to R3.5 regarding predictions about NK cells. We wish to clarify that we would not have been surprised at all to find that mosquitoes were a more sensitive assay for virus than our tissue-culture based assays- what is surprising is that monkeys for which we measured relatively high levels of DENV in our tissue culture assay did not infect mosquitoes, whereas monkeys with lower levels of virus did. We have added some additional interpretation of this finding based on a very recent paper by Lambrechts et al.¹⁰ in lines 379-389.

Minor things

C4.8. p.6 "Sylvatic ZIKV replicated to high levels in squirrel monkeys, stimulating low

neutralizing antibody titers" Stimulating? I don't get this. Isn't this more indicative of low antibody titers?

R4.8. We agree that the word stimulate may be a little jarring when referring to a lower level of antibody, but we have chosen to retain it for consistency with the rest of the manuscript.

C4.9. - p.7 Two squirrel monkeys had to be euthanised? I agree that one cannot attribute the reasons for this to them being infected with Zika with certainty, but one cannot exclude it either! Zika is known to cause neurological symptoms and it may cause arthritis. Whatever the cause, a mortality rate of 2/30 in infected animals (if I'm correct) is significant.

R4.9. As we report in the supplement (Text S.2), one of the two animals that had to be euthanized had a sore on her paw that did not heal because she continued to worry at it, and we are confident that this has nothing to do with ZIKV infection. As for the other animal that had to be euthanized due to seizures that we attribute to a brain lesion, we genuinely cannot be sure whether this is due to ZIKV or not, and we have tried to convey that in the manuscript.

- 1 Salazar, M. I., Richardson, J. H., Sanchez-Vargas, I., Olson, K. E. & Beaty, B. J. Dengue virus type 2: replication and tropisms in orally infected *Aedes aegypti* mosquitoes. *BMC Microbiol* **7**, 9 (2007). <https://doi.org:10.1186/1471-2180-7-9>
- 2 Azar, S. R. *et al.* Differential Vector Competency of *Aedes albopictus* Populations from the Americas for Zika Virus. *Am J Trop Med Hyg* **97**, 330-339 (2017). <https://doi.org:10.4269/ajtmh.16-0969>
- 3 Roundy, C. M. *et al.* Variation in *Aedes aegypti* Mosquito Competence for Zika Virus Transmission. *Emerg Infect Dis* **23**, 625-632 (2017). <https://doi.org:10.3201/eid2304.161484>
- 4 Weger-Lucarelli, J. *et al.* Vector Competence of American Mosquitoes for Three Strains of Zika Virus. *PLoS Negl Trop Dis* **10**, e0005101 (2016). <https://doi.org:10.1371/journal.pntd.0005101>
- 5 Vasilakis, N. *et al.* Antigenic relationships between sylvatic and endemic dengue viruses. *Am J Trop Med Hyg* **79**, 128-132 (2008).
- 6 Althouse, B. M. *et al.* Viral kinetics of primary dengue virus infection in non-human primates: a systematic review and individual pooled analysis. *Virology* **452-453**, 237-246 (2014). <https://doi.org:10.1016/j.virol.2014.01.015>
- 7 Althouse, B. M. & Hanley, K. A. The tortoise or the hare? Impacts of within-host dynamics on transmission success of arthropod-borne viruses. *Philos Trans R Soc Lond B Biol Sci* **370** (2015). <https://doi.org:10.1098/rstb.2014.0299>
- 8 Hanley, K. A., Azar, S. R., Campos, R. K., Vasilakis, N. & Rossi, S. L. Support for the Transmission-Clearance Trade-Off Hypothesis from a Study of Zika Virus Delivered by Mosquito Bite to Mice. *Viruses* **11** (2019). <https://doi.org:10.3390/v11111072>
- 9 O'Sullivan, T. E., Sun, J. C. & Lanier, L. L. Natural Killer Cell Memory. *Immunity* **43**, 634-645 (2015). <https://doi.org:10.1016/j.immuni.2015.09.013>
- 10 Lambrechts, L. *et al.* Direct mosquito feedings on dengue-2 virus-infected people reveal dynamics of human infectiousness. *PLoS Negl Trop Dis* **17**, e0011593 (2023). <https://doi.org:10.1371/journal.pntd.0011593>

REVIEWERS' COMMENTS

Reviewer #1 (Remarks to the Author):

The authors diligently implemented the majority of the suggested modifications proposed during the initial peer-review process for this manuscript. In instances where certain changes were not incorporated, a comprehensive explanation was provided. These revisions have markedly bolstered the credibility and scientific integrity of the manuscript. As a result, readers can now comprehensively grasp the scientific methodologies employed in this study and interpret the findings with accuracy, devoid of bias or incomplete information.

I do have one minor quibble. The authors state in their response letter that sequence authentication of challenge stocks is routine practice in their laboratories. However, I see no reference to authentication in the methods of the manuscript. Ideally, the authors's would include how the challenge virus sequence differs from the Genbank record and include the accession number for where these data are located (SRA or similar).

Reviewer #2 (Remarks to the Author):

The authors provided reasonable modifications to the MS text in response to my specific questions and comments. Furthermore, their speculation on more conceptual questions was appreciated.

Reviewer #3 (Remarks to the Author):

First off, i commend the authors for substantially improving the manuscript, and i thank them for taking the time to respond to my comments. Although the study still suffers somewhat from low sample sizes in some of the experiments, the revised submission is far more focused and is nicely put together. i do have one comment that i would consider minor; in the first submission, i felt the manuscript lacked a take-home message as to the point of the paper. The manuscript is much improved in this regard, but now the abstract leaves the reader hanging as to what the data collected actually mean. a simple wrap-up, or summary sentence would suffice. Journal word limits may be the issue, but this would greatly help the reader if at all possible.

Reviewer #4 (Remarks to the Author):

The response to the reviewers addressed most of the issues I had (I learned quite a lot from the other reviewers' comments, by the way), and the ms now reads much easier. The only issue I still have is with the hypotheses that are tested. Typically, if an experiment doesn't address an either-or situation, we think of hypotheses as of logically orthogonal (at least I do): different things may or not be true independently. Here, Hypotheses 2 and 3 are more special cases of Hypothesis 1. If there is no replication-clearance trade-off (Hypothesis 1) the other two cannot be true automatically. Vice versa, evidence for Hypotheses 2 and 3 (NK cells play a role, different trade-offs in different host species, respectively) automatically adds support for hypothesis 1. I see Hypothesis 1 as a fundamental one, Hypothesis 2 dealing more with mechanism and Hypothesis 3 exploring the evolutionary context. I'd encourage the authors to clarify at least the relationships between their hypotheses, and point out that they are not independent.

Finally the fact that viruses have lower net transmission from novel hosts is not indicative of them being subject to different trade-offs. Trade-offs such as the replication-clearance trade-off as studied here result from viruses hitting the constraints they are subject to. If the virus is unadapted, it may simply not have had the time yet to had to reach the constraint, which may actually be similar to the constraint limiting transmission (i.e., trade-off) in a well-adapted virus. Pointing this out may avoid some potential confusion.

Please see below for a summary of our responses to the final reviews from reviewers.

Reviewer #1 (Remarks to the Author):

The authors diligently implemented the majority of the suggested modifications proposed during the initial peer-review process for this manuscript. In instances where certain changes were not incorporated, a comprehensive explanation was provided. These revisions have markedly bolstered the credibility and scientific integrity of the manuscript. As a result, readers can now comprehensively grasp the scientific methodologies employed in this study and interpret the findings with accuracy, devoid of bias or incomplete information.

I do have one minor quibble. The authors state in their response letter that sequence authentication of challenge stocks is routine practice in their laboratories. However, I see no reference to authentication in the methods of the manuscript. Ideally, the authors's would include how the challenge virus sequence differs from the Genbank record and include the accession number for where these data are located (SRA or similar).

Response 1: Please see the additional text in Methods and quoted below:

“The DENV-2 stock was 3 C6/36 passages removed from a stock sequenced by Vasilakis et al.⁷³ (Genbank ID KU95559.1), and the ZIKV stock was 3 C6/36 passages removed from the stock sequenced by Ladner et al.⁷⁴ (Genbank ID EF105379.1). A diagnostic region of the envelope gene of each working stock of virus (nucleotides 1435 – 1744 for DENV-2 and 2163 – 2498 for ZIKV) were sequenced and showed no changes relative to the reference sequence.”

Reviewer #2 (Remarks to the Author):

The authors provided reasonable modifications to the MS text in response to my specific questions and comments. Furthermore, their speculation on more conceptual questions was appreciated.

Response: We thank the reviewer for their kind words.

Reviewer #3 (Remarks to the Author):

First off, i commend the authors for substantially improving the manuscript, and i thank them for taking the time to respond to my comments. Although the study still suffers somewhat from low sample sizes in some of the experiments, the revised submission is far more focused and is nicely put together. i do have one comment that i would consider minor; in the first submission, i felt the manuscript lacked a take-home message as to the point of the paper. The manuscript is much improved in this regard, but now the abstract leaves the reader hanging as to what the data collected actually mean. a simple wrap-up, or summary sentence would suffice. Journal word limits may be the issue, but this would greatly help the reader if at all possible.

Response 3: We have extensively revised the abstract in response to this review and suggestions of two scientists outside of the field, and we hope that this version is now satisfactory.

Reviewer #4 (Remarks to the Author):

The response to the reviewers addressed most of the issues I had (I learned quite a lot from the other reviewers' comments, by the way), and the ms now reads much easier. The only issue I

still have is with the hypotheses that are tested. Typically, if an experiment doesn't address an either-or situation, we think of hypotheses as of logically orthogonal (at least I do): different things may or not be true independently. Here, Hypotheses 2 and 3 are more special cases of Hypothesis 1. If there is no replication-clearance trade-off (Hypothesis 1) the other two cannot be true automatically. Vice versa, evidence for Hypotheses 2 and 3 (NK cells play a role, different trade-offs in different host species, respectively) automatically adds support for hypothesis 1. I see Hypothesis 1 as a fundamental one, Hypothesis 2 dealing more with mechanism and Hypothesis 3 exploring the evolutionary context. I'd encourage the authors to clarify at least the relationships between their hypotheses, and point out that they are not independent.

Finally the fact that viruses have lower net transmission from novel hosts is not indicative of them being subject to different trade-offs. Trade-offs such as the replication-clearance trade-off as studied here result from viruses hitting the constraints they are subject to. If the virus is unadapted, it may simply not have had the time yet to had to reach the constraint, which may actually be similar to the constraint limiting transmission (i.e., trade-off) in a well-adapted virus. Pointing this out may avoid some potential confusion.

Response 4: We appreciate the reviewer's clarification, and have reworded the last paragraph, as quoted below.

“In the current study, we investigated trade-offs between replication and clearance of sylvatic DENV and ZIKV, as well as specific immune responses associated with these trade-offs and potential differences in trade-offs between native and novel NHP hosts. This study tested three *a priori* hypotheses, of which the first is fundamental and the latter two consider mechanism and evolutionary context, respectively: Hypothesis 1 - sylvatic arboviruses experience a replication-clearance trade-off in both native and novel hosts, as evidenced by a positive relationship between the dose of virus delivered with peak virus titer, a positive relationship between virus titer and transmission to mosquitoes, and a negative relationship between peak virus titer and duration of viremia; Hypothesis 2 - NK cells can serve as a proxy for the innate immune responses that shape this trade-off, as evidenced by a negative relationship between NK cell mobilization immediately post-infection (π) and peak virus titer as well as levels of neutralizing antibody, and Hypothesis 3 - sylvatic arboviruses have achieved an optimal replication-clearance trade-offs in native hosts but have not had evolutionary opportunity to reach this optimum in novel hosts, resulting in less transmission from novel hosts.”